# Structural basis of the membrane intramolecular transacylase reaction responsible for lyso-form lipoprotein synthesis

Samir Olatunji [1], Katherine Bowen [2], Chia-Ying Huang [3], Dietmar Weichert [1], Warispreet Singh[4,5,6], Irina G. Tikhonova [4], Eoin M. Scanlan[2], Vincent Olieric [3] & Martin Caffrey [1]✉

Lipoproteins serve diverse functions in the bacterial cell and some are essential for survival. Some lipoproteins are adjuvants eliciting responses from the innate immune system of the host. The growing list of membrane enzymes responsible for lipoprotein synthesis includes the recently discovered lipoprotein intramolecular transacylase, Lit. Lit creates a lipoprotein that is less immunogenic, possibly enabling the bacteria to gain a foothold in the host by stealth. Here, we report the crystal structure of the Lit enzyme from *Bacillus cereus* and describe its mechanism of action. Lit consists of four transmembrane helices with an extracellular cap. Conserved residues map to the cap-membrane interface. They include two catalytic histidines that function to effect unimolecular transacylation. The reaction involves acyl transfer from the *sn*-2 position of the glyceryl moiety to the amino group on the N-terminal cysteine of the substrate via an 8-membered ring intermediate. Transacylation takes place in a confined aromatic residue-rich environment that likely evolved to bring distant moieties on the substrate into proximity and proper orientation for catalysis.

[1] Membrane Structural and Functional Biology Group, School of Medicine and School of Biochemistry and Immunology, Trinity College Dublin, Dublin, Ireland. [2] School of Chemistry, Trinity College Dublin, Dublin, Ireland. [3] Swiss Light Source, Paul Scherrer Institute, Villigen, Switzerland. [4] School of Pharmacy, Queen's University Belfast, Belfast, United Kingdom. [5] Present address: Faculty of Health and Life Sciences, Northumbria University, Newcastle upon Tyne, United Kingdom. [6] Present address: Hub for Biotechnology in Build Environment, Newcastle upon Tyne, United Kingdom. ✉email: martin.caffrey@tcd.ie

Aquarter of all proteins in *Escherichia coli* are lipoproteins[1]. Some of these are enzymes, others are inhibitors or components of complexes responsible for cell envelope biogenesis, nutrient uptake and detoxification. *E. coli*'s most abundant protein, Braun's lipoprotein, secures the outer membrane and the peptidoglycan layer together[2]. By virtue of being essential or virulence factors, the enzymes responsible for lipoprotein synthesis are important targets for the development of antibiotics to treat a growing number of drug resistant bacteria[3,4]. Relatedly, higher organisms have developed an alert system to respond to bacterial infections that is triggered by lipoproteins[5,6]. These are potent agonists of *Toll-like receptors* (TLRs) that, when activated, set in motion sequential innate and acquired immune responses eventually leading to the production of neutralising antibodies. In this capacity therefore, bacterial lipoproteins (BLPs) function as natural adjuvants[7,8]. Indeed, BLPs have been used as such in a number of vaccines[9,10]. They can also be included in vaccine formulations as potent antigens[11]. Clearly, lipoproteins deserve our attention for the myriad functions they serve in the life of the bacterial cell and their hallmark signature as 'other' or 'foreign' duly responded to by the immune system.

Until recently, the canonical pathway of BLP posttranslational processing in Gram-negative organisms included three enzymes (Fig. 1). All reside in the cytoplasmic membrane and all have active sites facing the periplasm. The first enzyme, lipoprotein diacylglyceryltransferase, Lgt, transfers a diacylglyceryl (DAG) moiety from a phospholipid, usually phosphatidylglycerol, PG, to a conserved cysteine in the membrane-anchored signal peptide of a preproBLP[12]. The lipid and the protein are in thioether linkage and the modification produces a proBLP that is now doubly anchored in the membrane. The second enzyme is lipoprotein signal peptidase II, LspA[13–16]. It removes the signal peptide from the proBLP by cleaving to the N-side of the cysteine and generates a DAG-modified BLP as product. This diacylated BLP (DA-BLP) is acted on by a third enzyme, lipoprotein *N*-acyltransferase, Lnt, which *N*-acylates the free amino group of the lipidated cysteine in the DA-BLP thereby generating a triacylated BLP (TA-BLP) product[17–20]. The *localisation of lipoprotein*, Lol, system of proteins then shuttles the TA-BLP to the outer membrane for directed partitioning between its inner and outer leaflets[21]. The source lipid for the Lnt reaction is predominantly phosphatidylethanolamine, PE. It is the acyl chain at the *sn*-1 position on the glyceryl moiety of PE that is transferred. While DA-BLPs activate the TLR2/TLR6 system, TA-BLPs are TLR2/TLR1 system agonists[6,22,23].

In 2012, Kurokawa et al.[24], identified three BLP variants based on mass spectrometric (MS) analyses of lipoproteins from Gram-positive firmicutes known to lack Lnt sequence orthologs. They included BLPs with an acetyl and a dipeptide in amide linkage to the N-terminal cysteine. The third variant was a lyso-BLP, so named because it has just one acyl chain in ester linkage to the glyceryl moiety attached to the N-terminal cysteine and a second acyl chain in amide linkage to its α-amino group (Fig. 1). It was proposed then that the enzyme responsible for lyso-BLP synthesis was either a transacylase which transfers one of the acyl chains from the DAG of DA-BLP to the α-amino group of the N-terminal cysteine. Alternatively, it was an *O*-deacylase which cleaves an acyl chain from the DAG of TA-BLP generated by an unidentified Lnt-like activity[24]. On the basis of an intergenic complementation rescue assay, the membrane integral enzyme responsible was identified in 2017 as likely having the former activity[22]. The enzyme was shown to be chromosomally encoded in low-GC Gram-positive bacteria and was named lipoprotein intramolecular transacylase, Lit. Which of the two acyl chains in the DAG that is transferred was not known although some data available at the time suggested it was the *sn*-2 chain. Nor indeed was the mechanism of transfer known. Speculation included an acylated enzyme intermediate and a direct transfer reaction. Work reported here, and recently by Armbruster et al[1]., favour the latter mechanism.

In 2019, Armbruster et al., identified a second *lit*, *lit2*, on a mobile genetic element in certain strains of the Gram-positive bacteria, *Listeria monocytogenes* and *Enterococcus* spp[23]. Expression of *lit2* was shown to be tightly regulated. It was induced by copper and was co-transcribed with a second *lgt* (*lgt2*) and several copper-resistance determinants. The original chromosome-encoded Lit, from here on referred to simply as Lit (*Bacillus cereus*, 218 residues), and Lit2 (*L. monocytogenes*, 204 residues) have analogous primary structures with sequence identities and homologies of 27 and 50%, respectively. They have similar enzymatic activities and their lyso-BLP products are considerably weaker ligands of the TLR systems than either DA-BLP or TA-BLP[22,23].

In the space of 8 years therefore, an additional class of cysteine-modified lipoproteins and the enzymes responsible for their synthesis have been identified. An intriguing link to copper homoeostasis in bacteria and to immune response attenuation raises interest in these lipoprotein processing enzymes.

In this work, we report the high-resolution crystal structure of Lit from *B. cereus* with a view to understanding how the enzyme

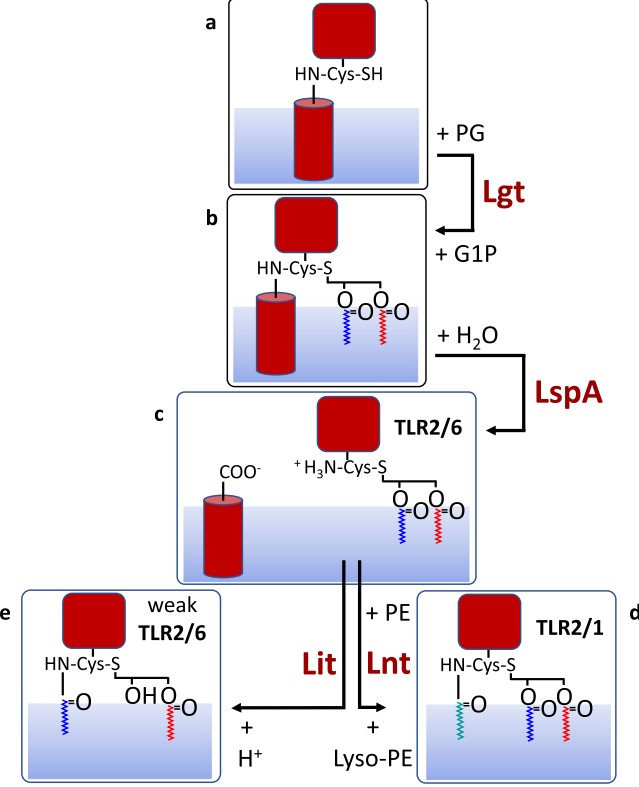

**Fig. 1 Bacterial lipoprotein cysteine-based post-translational modification pathways. a** Preprolipoprotein is converted to (**b**) prolipoprotein by the action of Lgt which uses PG as the lipid substrate. G1P is the second product of the reaction. Prolipoprotein is cleaved by the signal peptidase II, LspA, generating (**c**) apo-lipoprotein and signal peptide. In Gram-negative bacteria, Lnt *N*-acylates the apo-lipoprotein using PE forming (**d**) triacylated lipoprotein and lyso-PE. In low-GC% Gram-positive bacteria such as *B. cereus* and *E. faecalis*, Lit performs an intramolecular transacylation on apo-lipoprotein forming (**e**) lyso-lipoprotein. Lipoprotein interactions with receptors, TLR2/1 and TLR2/6, are indicated in **c**–**e**. A different type of lipoprotein, where processing involves signal peptidase I cleavage and acylation at an N-terminal glycine, has been described recently[39,40].

functions at a molecular level. Combined with a direct functional assay, analytical nuclear magnetic resonance (NMR) spectroscopy, mutagenesis, molecular dynamics simulations (MDS) and quantum mechanics/molecular mechanics (QM/MM), a model for how the enzyme carries out intramolecular transacylation emerges with insights into substrate selectivity.

## Results

**Lit structure.** *B. cereus* is a Gram-positive facultative anaerobic bacteria that lives on food and in soil[25]. In humans, it is responsible for foodborne illnesses that include severe diarrhoea and vomiting. It is used as a probiotic in the diet of chickens, pigs and rabbits. A hexahistidine N-terminally tagged variant of Lit from *B. cereus* was over-expressed in *E. coli* and was crystallised by the in meso method with monoolein as the host lipid[26,27]. Two structures were solved; one at 2.27 Å, the other at 1.95 Å (Supplementary Table 1). The former, in space group *P*2₁, displayed type I crystal packing (Supplementary Fig. 1). The latter, in space group *P*2₁2₁2, had a packing that is less obviously type I that may have emerged as a result of a polymorphic transition from a nascent crystal with type I packing[28]. Both structures have two similar molecules in the asymmetric unit. Root mean-square deviations calculated for $C_\alpha$ atoms in the different structures (211 to 216 residues) range from 0.21 to 0.40 Å. Here, we focus on molecule B (MolB) in the 1.95 Å structure, which has the better quality electron density.

The protein assumes the shape of a truncated cone that spans the membrane with four transmembrane helices (M1–M4) (Fig. 2). Its wider end extends into the extracellular space. The narrower part is embedded in the membrane. Very little of the protein, that includes both N- and C-termini, extends into the cytoplasm. Consistent with the positive inside rule, this end of the protein is richly cationic. The protein has a pseudosymmetry between its N- and C-terminal halves. The N-terminal half consists of a membrane domain (MD1) and an extracellular globular domain (EGD1). MD1 includes two membrane helices (M1 and M2). M1 and M2 are linked by EGD1 consisting of three helices (H1–H3) and three turns of the M2 helix that extend beyond the membrane. The corresponding parts of the C-terminal half include a membrane domain (MD2) with two membrane helices (M3 and M4), and a globular domain (EGD2) with two helices (H4 and H5) and two turns of the M4 helix that reside in the extracytoplasmic space. H1 and H4 are amphiphilic and are expected to sit, in part or entirely, on the membrane surface. Monoolein molecules from the mesophase in which crystallogenesis occurs coat the protein surface recapitulating the native membrane and occupy a volume inside the protein. We assume these internal lipids take the place of lipoprotein substrate and product acyl chains, as discussed below. The pseudosymmetry between the structure of the N- and C-terminal halves of Lit is not obvious in their sequences.

The two central helices, M2 and M3, of the membrane-embedded domain are connected at the intracellular side by a short loop. They splay apart in the shape of a V as they cross the membrane. The N- and C-terminal membrane helices, M1 and M4, straddle the base of the V. This results in the four TM helices forming a tight bundle toward the intracellular side of the membrane. Viewed from the cytoplasm, these four helices are arranged quite symmetrically about the loop connecting M2 and M3 (Supplementary Fig. 2). As the four TM helices cross the membrane, they fan out forming a hemi-cylinder, the concave surface of which is exposed to the membrane interior and is decorated with hydrophobic residues. The hemi-cylinder, which roughly resembles a four fingered hand, has helices, M3, M4, M1 and M2 (fingers on the hand) arranged from left to right in that

order (Fig. 2). The fingertips of M2 and M4 extend out of the membrane into the extracellular space. The extracellular end of the hemi-cylinder is covered by a dome-shaped cap domain (CD) which, in turn, is composed of EGD1 and EGD2. The shape complementarity between EGD1 (groove, concave) and EGD2 (bulge, convex) where they meet is striking. EGD1 creates a large concave pocket into which sits the convex shaped EGD2. Interactions between the two are predominantly hydrophobic. The dome-shaped CD has a hydrophobic interior surface. It extends away from the fan of transmembrane helices with its convex apolar surface facing down toward the membrane. In so doing, it creates an opening from the membrane into the apolar dome interior below the amphiphilic H4. Based on a Dali search[29], the structural fold observed in Lit has not been reported previously. It is stable in a membrane as judged by MD simulations (Supplementary Fig. 3 and Supplementary Movie 1). A homology model of Lit2 from *L. monocytogenes* is remarkably similar to the template Lit crystal structure from *B. cereus* (Supplementary Fig. 4).

**Putative active site.** Sequence analysis identified several highly conserved residues in Lit that include His85, Val89, Phe149, His153, Phe157 and Trp162 (Supplementary Table 2 and Supplementary Fig. 5). Conserved residues have stood the test of evolutionary time because they serve a role in catalysis, in folding and/or structural stability. They all map onto the dome-shaped structure that sits atop the transmembrane helices in the extra-cytoplasmic space of the apoprotein (Fig. 3). We assume therefore that this is where the active site of the protein resides. This makes functional sense given that this same orientation, with the active site toward the extracytoplasmic interface of the membrane, is adopted by the three other lipoprotein processing enzymes, Lgt, LspA and Lnt[15,18,30]. It is also where the substrate's DAG-modified cysteine is expected to situate[15]. Notably, two structured monoolein molecules that derive from the crystallisation process reside with their glyceryl head groups coordinated with the two conserved histidines (Fig. 3 and Supplementary Fig. 6). These lipids are considered surrogates of the DAG moiety of the DA-BLP substrate and serve to reinforce the view that they sit proximal to the active site of Lit. Of the conserved residues, His85 and His153 are the only ones with polar and potentially charged side chains of the type typically found as catalytic residues[31]. They are within 5 Å of one another and are situated in the protein at the level of or slightly above the membrane interface. They are surrounded by aromatics (Phe86, Phe145, Phe149, Phe152, Phe157, Trp162, Phe164 and Phe180) several of which are highly conserved (Supplementary Fig. 7, Supplementary Table 2 and Supplementary Movie 1). Thus, stabilising π-stacking as well as cation–π interactions between the two histidines and neighbouring aromatics is possible regardless of the protonation state of either or both histidines[32]. For now, we assume His85 and His153 constitute the catalytic dyad in Lit that functions as a monomer. Evidence supporting this view comes from MD and QM/MM simulations and from structure-based site-directed mutagenesis, described below. An evaluation of the mutagenesis work was enabled by the development of a direct method for quantifying Lit transacylase activity.

**Transacylase assay.** Lit catalyses a transacylation reaction in which there is but one lipoprotein substrate and one lipoprotein product. The molecular changes that occur during the reaction include the conversion of an ester and an ammonium group to a hydroxyl group and an amide (Fig. 1). We reasoned that the slight polarity difference between the substrate and product might be exploited such that they could be separated and quantified by thin

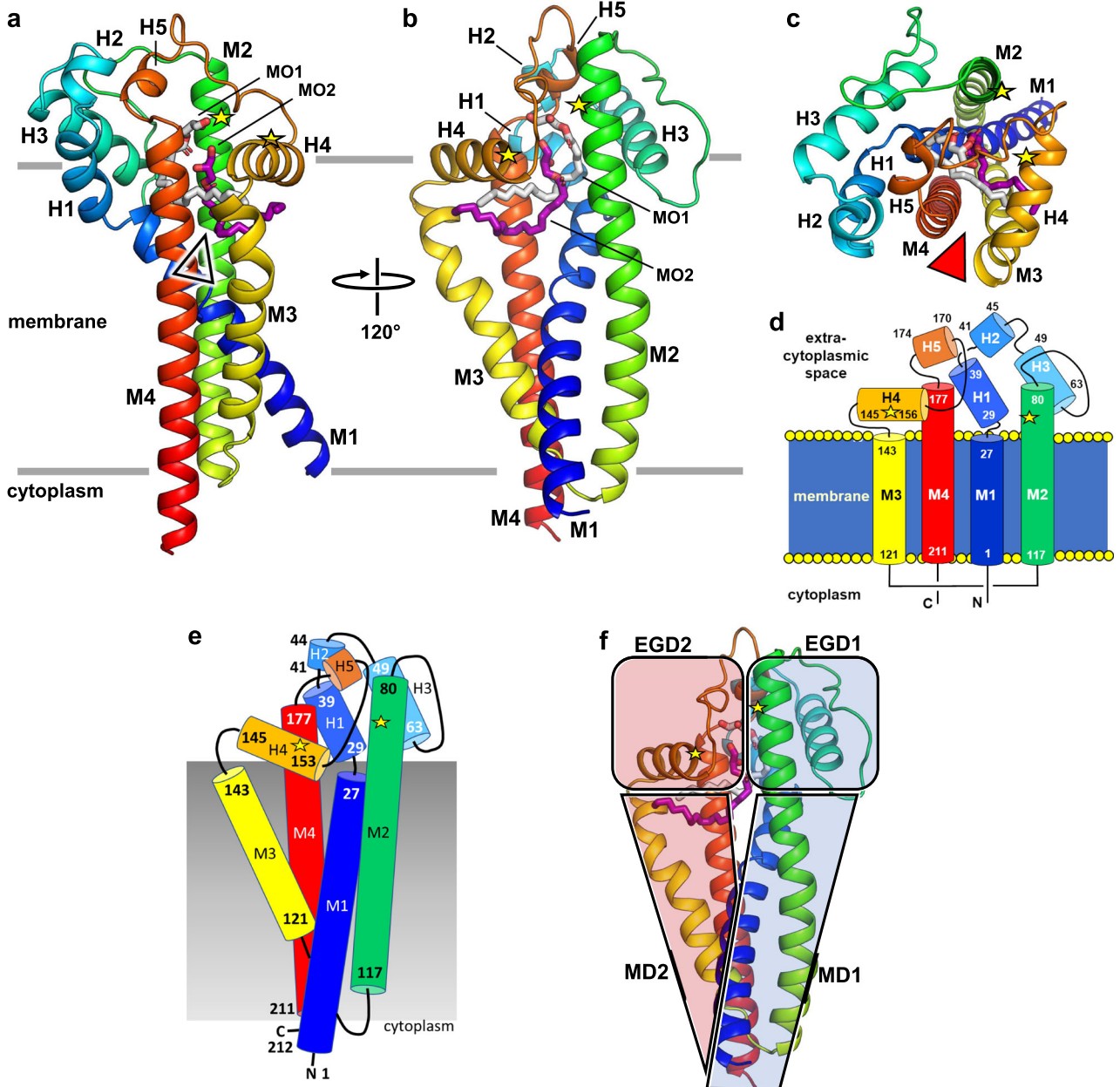

**Fig. 2 Overall architecture of apo-Lit from *B. cereus*. a** View from the membrane plane into the active site from the front. Substrate and product putatively enter and leave the active site via a fenestration (arrowhead) that opens between the extracyctoplasmic halves of M3 and M4. Structured lipids in the active site pocket have grey and purple carbon atoms for clarity. Approximate location of catalytic residues, His85 and His153, are marked with yellow stars on M2 and H4, respectively. Monoolein molecules that coat the surface of the protein have been omitted for clarity. Approximate location of the membrane boundaries are shown as horizontal lines. **b** View from the membrane plane into the active site from the back. **c** View from the extracellular space into the active site. Fenestration marked with an arrowhead. **d** Topology of Lit in a simplified representation to highlight the elements of symmetry in the protein. **e** A view of the protein from the back with helices shown as rods. **f** The pseudo twofold structural symmetry in Lit, illustrated by dividing the protein into N- and C-terminal halves (transparent blue and red), can be seen by comparing **e** and **f** which are in the same orientation. MD and EGD correspond to the membrane domain and the extracellular globular domain, respectively.

layer chromatography (TLC) provided the molecular weights involved were not too great. To facilitate visualisation, the lipoprotein substrate was labelled with a fluorophore (FP2) (Supplementary Fig. 8). FP2 has a DAG-modified cysteine at the N-terminus followed by two serines and a C-terminal lysine with the fluorophore, 2-aminobenzoyl (Abz), in amide linkage to its ε-amino group. By optimising TLC mobile phases, one was identified that cleanly separated the Lit substrate and product. Both are visible by fluorescence on the TLC plate and can be quantified by image analysis. NMR spectroscopy was used to

identify the product and to show that the acyl chain at the *sn*-2 position on the glyceryl of the substrate is transferred in the Lit reaction (vide infra).

This simple TLC assay based on FP2 provides a convenient and direct method for quantifying Lit activity. With it, conditions for performing activity-based assays were established (Supplementary Figs. 9 and 10). The enzyme was shown to be inactive with the (*S*)-stereoisomer of the DAG in the substrate and, by extension, specific for the (*R*)-stereoisomer. It was optimally active in the pH range from 4.4 to 5.4. Subsequent assays were

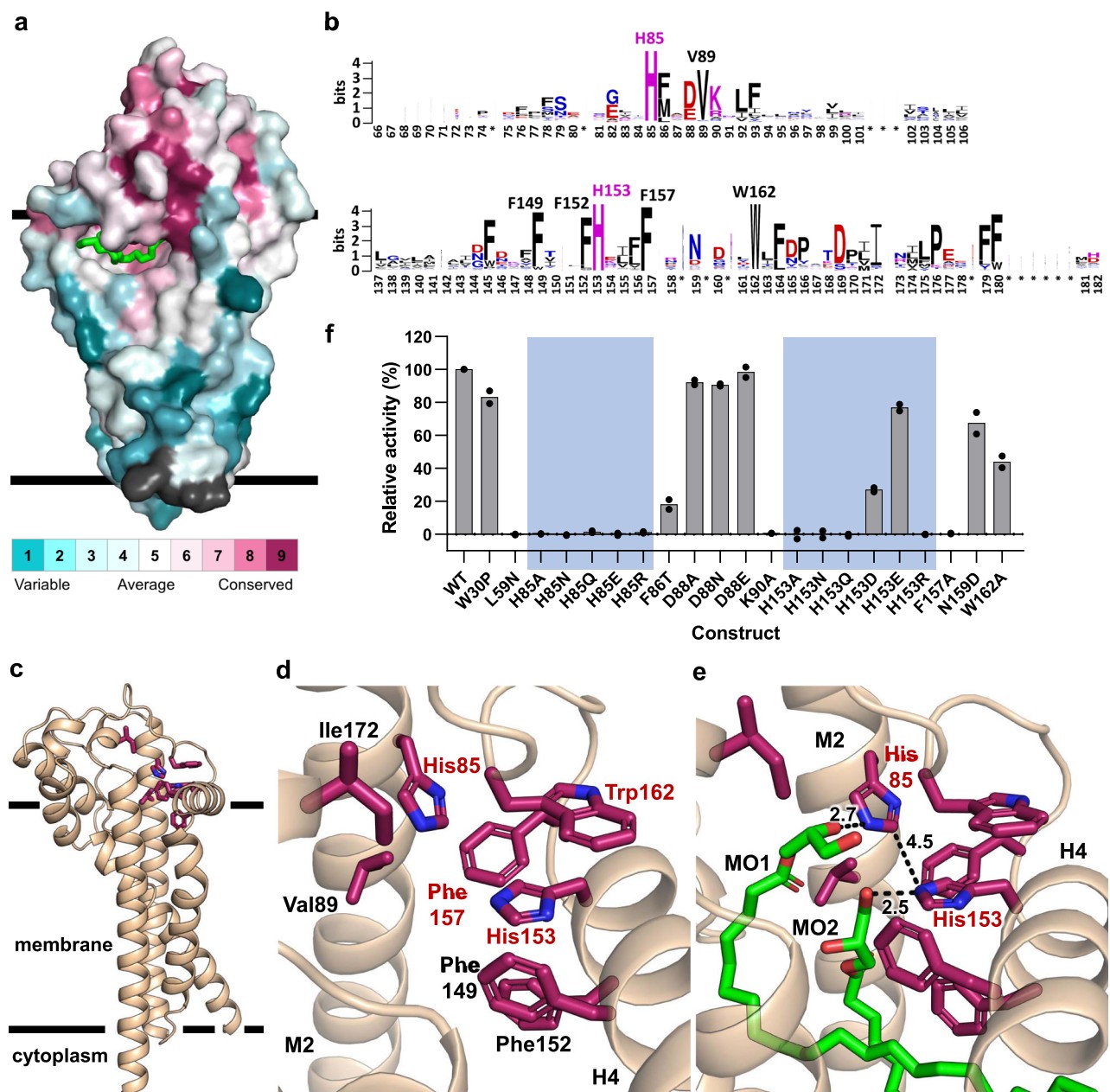

**Fig. 3 Location and mutational analysis of conserved residues demark putative active and substrate binding sites in Lit. a** Structure of Lit with conserved residues colour coded from blue to purple corresponding to low and high % conservation, respectively. **b** Sequence logo[74] representation of a Clustal Omega[71] sequence alignment of 788 Lit orthologs from the UniProt Reference Proteomes database[68]. **c** View from the membrane at the eight conserved residues (purple carbons) in Lit that are assumed to define the substrate-binding pocket. The pocket resides at and above the surface of the membrane (black bars). Conserved residues are at ≥90% conservation. **d** A zoomed in view of the conserved residues in Lit from which internal lipids have been removed. Red lettering denotes highly conserved residues (≥98% conservation). **e** A zoomed in view of the conserved residues in Lit along with the structured monoolein molecules that co-crystallise with the enzyme. Electron densities that define the lipid molecules are shown in Supplementary Fig. 6. The lipids are considered surrogates for the diacylglyceryl chains in the lipoprotein substrate. Distances shown as dashed lines are in ångströms. **f** Transferase activity of Lit mutants. Values are expressed as a percentage of wild-type (WT). A TLC assay was used to quantify Lit activity with FP2 as substrate on the basis of a 30 min incubation at 37 °C. The standard deviation is calculated on the basis of duplicate measurements. Mutants of the putative catalytic residues H85 and H153 are highlighted for clarity.

usually performed at 37 °C for 30 min in a buffer containing 25 mM MES/NaOH, pH 5.4, 0.2 M sodium chloride, 0.02%(w/v) LMNG, 0.15 mM FP2 and 10 μM Lit. Under these conditions, the enzyme had an apparent turnover number of ~0.25 mole of substrate per min per mole of Lit.

As part of the characterisation of Lit as an enzyme, its sensitivity to divalent cations was examined (Supplementary Fig. 10). To begin with, activity was determined in the presence of

EDTA which was shown to have no effect. This suggested that cations, at inhibitory concentrations, were not present in the enzyme preparation and that divalent cations may not be required for catalysis. Subsequent assays were performed in the presence of calcium, magnesium, zinc and copper. Neither calcium nor magnesium had any effect on activity. However, conversion of substrate to product was reduced considerably by zinc and was stopped completely by copper.

Given that copper is implicated in Lit2 upregulation[23], we further investigated the effect of copper on Lit. It seemed plausible that copper could inhibit the enzyme given that its putative active site residues include a pair of proximal histidines which might bind copper. An alternative hypothesis was also considered that copper might coordinate with the lipopeptide substrate[23] in which form it is no longer processed by Lit. Thus, assays were performed at increasing concentrations of copper (0–1 mM) under conditions where the enzyme (0.01 mM) and substrate (0.15 mM) concentrations were held constant (Supplementary Fig. 10c, d). Copper had no effect on activity until its concentration rose to 0.05 mM and above. No activity was observed at 0.3 mM copper and above. These results suggest that copper does not affect the enzyme directly. Rather, that it binds to the substrate, which in complex with copper is no longer converted to product. It has been proposed that BLPs of form DA-BLP, by virtue of having an amino group and sulphur (and oxygen) atoms in close proximity in the terminal cysteine, can directly bind copper[23]. Exploratory NMR measurements show that copper does indeed interact with the N-terminus of the lipopeptide substrate (Supplementary Figs. 11–14).

**Evaluating the role of select residues in Lit action**. With the transacylation assay in place and optimised, the proposed mechanism of Lit action with His85 and His153 as the catalytic dyad and the role of other conserved residues were tested using purified mutant enzyme (Fig. 3f). That His85 and His153 play a role in catalysis was evaluated by preparing mutants from His to Ala, Asn, Asp, Gln, Glu and Arg at each position. Of the 12 mutants, only His85Asp failed to express. The remaining five His85 mutants expressed protein, however they were without significant activity consistent with this residue playing a role in catalysis. Four of the His153 mutants (Ala, Asn, Gln and Arg) likewise were inactive. However, His153Asp and His153Glu had 27 and 77% of wild-type activity, respectively. These results suggest that His153 is the second residue that functions alongside His85 in a general acid–base transacylase reaction mechanism, as proposed[1]. Thus, His85 would remain protonated throughout the reaction while His153 (and His153Asp and His153Glu) would be free to engage in protonic equilibrium. As discussed below, His85Ala and His85Arg were crystallised in the presence of the FP2 substrate and structures of both mutant forms were obtained. These observations indicate that His85 is not essential for the expression, folding or stability of Lit. That His85 remains fully protonated through the reaction is supported by its observed hydrogen bonding with structured lipid MO1 and with structured Water1 that, in turn, interacts with Asp88 and Asn159 (Supplementary Fig. 7). QM/MM data also support this conclusion (Supplementary Figs. 15 and 16).

Phe157 is the most highly conserved residue in Lit. It sits on the loop connecting H4 and H5 facing into the active site equidistant from and just below catalytic His85 and His153 (Supplementary Fig. 7a). Given that Phe157 is essentially invariant, the fact that the Phe157Ala construct is inactive comes as no surprise. However, it is not clear what role this residue plays. It may be that the charge distribution and π electrons in its phenyl ring interact functionally with nearby His85 and His153 (and presumably with His153Asp and His153Glu)[32] to facilitate the Lit reaction. It may also play a role in orienting the substrate optimally for binding and for reaction. The fact that it expressed well and behaved as normal during purification suggests that Phe157 is not essential for folding or stability. MD simulations and QM/MM calculations show that Phe157, along with other local aromatic residues, through interactions with catalytic histidines and with substrate, intermediates and product, likely plays a role in facilitating the transacylation reaction (Supplementary Tables 3 and 4).

Lys90 sits on M2 at the level of the extracytoplasmic membrane facing away from the active site (Supplementary Fig. 17). In orthologs, arginine often takes its place suggesting that a cationic side chain is important at this position. In Lit, Lys90 interacts closely with backbone carbonyls in the turn just after amphiphilic H3 and with the helix dipole in H3. It likely contributes to stabilising part of the extramembrane dome above and to one side of the active site. Not unexpectedly, the Lys90Ala mutant is essentially inactive (Fig. 3f).

We have speculated that the extramembrane domain, with its internal apolar surface, is an important functional feature of Lit. We tested this hypothesis by mutating Leu59 and Phe86, both of which contribute to the hydrophobic surface, to polar residues. Leu59Asn was inactive while Phe86Thr retained just 18% of wild-type activity. Both substitutions were with residues approximately equal in size to the original but they differed in polarity. The results are consistent with the need for a dome with an apolar inner surface for full activity. Altering this feature likely adversely affects how the substrate engages with the enzyme.

This targeted mutational study reinforces the view that the active site of the enzyme is made up of the highly conserved residues His85, His153 and Phe157 and that the apolar under-surface of the CD is an important feature of the substrate-binding pocket.

**Lipoprotein substrate binding**. The lipoprotein substrate and product of Lit have two acyl chains attached to an N-terminal cysteine. In the substrate, the chains are next to one another esterified to neighbouring hydroxyls on the glycerol backbone. In the product, one chain is on the glycerol while the other is attached to the α-amino group of the cysteine. Presumably, these two quite distinct arrangements of acyl chains on the cysteine are accommodated in the active site pocket of the enzyme. How this might happen is suggested by the way in which two structured lipid molecules sit in the putative binding pocket of the apoenzyme (Fig. 3). These are monooleins that derive from the host mesophase used for crystallogenesis.

The two monooleins reside with their head groups below the interior apolar surface of the dome domain on the extracytoplasmic side of the membrane (Supplementary Fig. 18). MO1 sits higher up in the dome with its head group above the plane of the membrane interface. MO2 is seated lower down; its head group sits at about the level of the membrane interface next to MO1. The sn-2 hydroxyl of MO1 coordinates with His85. The sn-3 hydroxyl of MO2 coordinates with His85 and His153. Their acyl chains extend away from the dome into the apolar recesses of the membrane in various ways reflecting the flexible nature of hydrocarbon chains. In MolB of the asymmetric unit, the chains run parallel to one another below the apolar surface of amphiphilic H4. In MolA, they splay apart with one below H4 as in MolA and the other extending out of the binding pocket to contact M3 within the membrane. Because the two acyl chains originate as ester linkages some distance from one another in the two monooleins, our interpretation is that these structured lipids more closely resemble the acyl chains in the Lit lipoprotein product than they do its substrate. However, we have used the orientation and conformation of MO1 and MO2 to provide a basis for how the lipoprotein substrate and product might be placed in the binding pocket for reaction and for use in the computational studies described below (Supplementary Fig. 19).

The primary hydroxyl of MO1 extends toward and can be seen through the opening in the dome of apo-Lit. It coordinates with a structured water (Wat2) that sits in the centre of the opening

(Supplementary Fig. 18). This suggests that the peptide tether between the diacylated cysteine and the folded domain of the lipoprotein may well extend through this same opening. The dome has a hemispherical shape and an apolar internal surface. Thus, any entity such as a diacylated cysteine to which is attached a polar peptide tether, that enters the dome as proposed, will likely rise within the dome to position its polar parts above the level of the membrane interface. This presumably would place the sn-2 oxygen of the lipoprotein substrate's glyceryl moiety next to the catalytic histidine dyad for reaction. Because the enclosing apolar surface has a concave shape, those parts of the diacylated cysteine that might normally be distant in a less confining space will be cinched together to become proximal within the dome. Thus, the dome may have evolved to bring the donor sn-2 ester moiety of the DAG and the acceptor α-amino group on the cysteine into close proximity and alignment for reaction. The (R, R) stereochemistry at the DAG-modified cysteine undoubtedly facilitates optimal substrate presentation also.

With a view to establishing how Lit interacts with its substrate, attempts were made to obtain a crystal structure of an inactive form of the enzyme bound to the lipopeptide substrate, FP2, used for acyltransferase assay work. The His85Ala and His85Arg mutants were chosen for complex formation since both were essentially inactive with this substrate. The cubicon method[33] was used to form the complex in the cubic mesophase as a prelude to crystallogenesis of His85Ala. Crystals and structures were obtained with both mutants. However, while density that could possibly be identified with the substrate was observed with the two mutants, it was not reliably fit with the lipopeptide as model (Supplementary Fig. 20). Our conclusion therefore is that the density observed in the substrate-binding pocket of these two constructs corresponds to structured monoolein molecules.

In the apo form of Lit, M3 is bent midway along its length between Pro134 and Pro140 (Fig. 3). At the front of the bend that faces into the membrane is a structured water, Wat89. This water molecule is coordinated by backbone carbonyls of Ile135 and Ala136 and by the backbone amide of Leu139 with reasonable tetrahedral interaction geometry (107.7°, 83.0° and 117.2°). Thus, Wat89 provides compensating hydrogen bonds broken as a result of the bend and, in so doing, stabilises the bend. In the His85Ala and His85Arg mutant structures obtained with crystals grown in the presence of the lipopeptide substrate, a different kink appears in M3 that causes it to curve away from M4 (Supplementary Fig. 20c). Deviation from a straight helix is no doubt aided by Gly126, Pro134 and Pro140 in this helix. Interestingly, a water molecule, Wat2 in His85Ala, stabilises the kink through hydrogen bonds with the backbone carbonyls of Ala14 and Thr130 and the amide of Phe18. Curvature in M3 is such that it separates from M4 toward its extracellular end to a greater extent than seen in wild-type Lit (Supplementary Fig. 21). In so doing, it acts as a gate to create an opening between the two helices that leads into the putative active site. The fenestration extends into the dome providing a conduit at least 10 Å wide along its length presumably for both the polar peptide and apolar lipid parts of the lipopeptide substrate and product to move into and out of the binding pocket. MD simulations support this view (Supplementary Fig. 22 and Supplementary Movies 2–4).

The particular 'open' conformations captured in the inactive constructs likely reflect the fact that the catalytic residue, His85, has been mutated to non-conservative residues (Supplementary Fig. 21). These Ala for His substitutions interfere with the tight packing and strong π–cation and π–π interactions of conserved aromatic residues in and around the active site of the native enzyme. Indeed, in both mutants aromatics that include Phe149, Phe152, Phe157, His153 and Trp162 have rearranged in a manner that may facilitate opening the M3 gate. Thus, while these mutant forms are not physiologically relevant, the structures they adopt may reveal features, such as the open gate, that are functionally relevant.

**Reaction mechanism.** One of the goals of this investigation was to establish the mechanism of the Lit reaction responsible for the production of lyso-BLPs in low-GC Gram-positive firmicute bacteria. Two mechanisms have been proposed[1,24] (Supplementary Fig. 15). One involves transfer of an acyl chain from the DAG of DA-BLP to the enzyme and thence to the free N-terminal amino group to form lyso-BLP. The other is an intramolecular shift of the chain to the free amino group of DA-BLP. To date, no direct evidence for either of these mechanisms has been provided. Indeed, until very recently, which of the two acyl chains on the DAG that is transferred remained unknown. These and related mechanistic issues were addressed in the following studies.

Glycerophospholipids are synthesised in bacteria beginning with the (R)-enantiomer of glycerol-3-phosphate[34]. Thus, the DAG in lipoproteins is expected to be of the (R)-stereochemical form. This hypothesis was tested in the current study with stereochemically pure (R)- and (S)-lipopeptide substrates as used in the transacylase assay (Supplementary Fig. 8). Lit showed no product formation after extensive incubation with the (S)-lipopeptide. However, with the (R)-form the enzyme converted ~70% of the substrate to product in 30 min at 37 °C (Supplementary Fig. 9b–d). This result shows that the enzyme is stereoselective and is specific for the (R)-lipopeptide substrate.

To establish which of the two chains is transferred in the course of the Lit reaction, a synthetic lipopeptide substrate incorporating a mixed chain DAG was used. Specifically, protiated and perdeuterated acyl chains were positioned, respectively, at the sn-1 and sn-2 positions of the glyceryl moiety (Supplementary Fig. 8). Hereafter, the mixed chain substrate is referred to as dFP2. Apart from being deuterated, dFP2 is identical to the protiated FP2 substrate used for all other work, including activity assays, reported in this study. The enzyme was incubated with dFP2 under standard assay conditions and the reaction was allowed to proceed for 18 h to maximise product (lyso-dFP2) formation. Substrate and product were separated by preparative TLC and the product band was collected, extracted and the extract used for NMR (Supplementary Figs. 23–26).

An analysis of the $^1$H−$^{13}$C heteronuclear single quantum coherence (HSQC) and total correlation spectroscopy (TOCSY) NMR data (Supplementary Figs. 27–34) show that the oxygen at C2 of the glyceryl moiety in the monoacylglyceryl of the lyso-dFP2 product is no longer acylated and exists as a secondary hydroxyl. At the same time, the amide in the lyso-dFP2 product links the amino group of cysteine to the perdeuterated acyl chain. Further, measurements with lyso-FP2 show that the sn-2 position was no longer acylated and that a protiated acyl chain was part of the amide linkage in the lyso-FP2 product. These data prove that the acyl chain transferred by Lit comes from the sn-2 position of the DAG in the BLP substrate. A similar conclusion was reached by Armbruster et al[1]. on the basis of MS analysis of a recombinantly produced lipoprotein containing deuterated acyl chains in the DAG moiety.

To further explore the reaction catalysed by Lit, QM/MM calculations were performed to evaluate proposed transacylation mechanisms. The first calculation considered the nucleophilic mechanism which involves formation of a transient acylated enzyme intermediate (Supplementary Fig. 15a). The simulation was set up with His85 and His153 as catalytic residues. To simulate the first step in the reaction, the epsilon nitrogen atom of His153 was moved closer to the carbonyl carbon atom of the substrate sn-2 chain to form the N-C bond of the acylated

intermediate. However, the potential energy scan showed no evidence of intermediates nor of transition states. In addition, the energy cost of the process was unrealistically high[35], in excess of 25 kcal/mol, which implies that the reaction is kinetically inaccessible (see Methods). For these reasons and because acyl-histidine intermediates are rare[36], the proposed mechanism was discounted. Indeed, the only time histidine dyads function as nucleophiles is in phosphotransferases, where the histidines interact directly with one another[31]. It might also be argued that acylating the enzyme creates a monoacylated lipopeptide and that a single chain may not be enough to keep the intermediate stably in place. If formed, it may partition out of the enzyme's binding pocket. Indeed, MD simulations show that the monoacylated lipopeptide interacts weakly with residues in the enzyme (Supplementary Table 3).

The alternative, general acid–base mechanism was evaluated by invoking His153 as the nucleophile. It abstracts a proton from the free α-ammonium group of the DAG-modified cysteine in the

DA-BLP substrate (Fig. 4, Supplementary Figs. 15b and 16, Supplementary Table 5 and Supplementary Movies 5–7). The newly formed amine attacks the carbonyl carbon at the *sn*-2 position of the DAG forming an eight-membered heterocyclic intermediate with a tetrahedral carbon and a tetrahedral nitrogen. The negative charge on the oxygen bonded to the tetrahedral carbon is stabilised by the proximal His85 which remains protonated throughout the reaction. A deprotonated His85 next to the cyclic intermediate does not allow the reaction to proceed (Supplementary Fig. 16). The positive charge on the heterocycle nitrogen repels the imidazolium side chain of His153 causing it to rotate away from the active site toward the aqueous phase where it is stabilised by π–cation interaction with conserved Trp162 (Supplementary Movie 6). Collapse of the cyclic intermediate gives rise to the uncharged lyso-BLP product with its resonance-stabilised *trans* amide bond. The amide linkage, with its affinity for hydrogen bond donors and acceptors, associates with two water molecules (Supplementary Fig. 35). The need to more fully

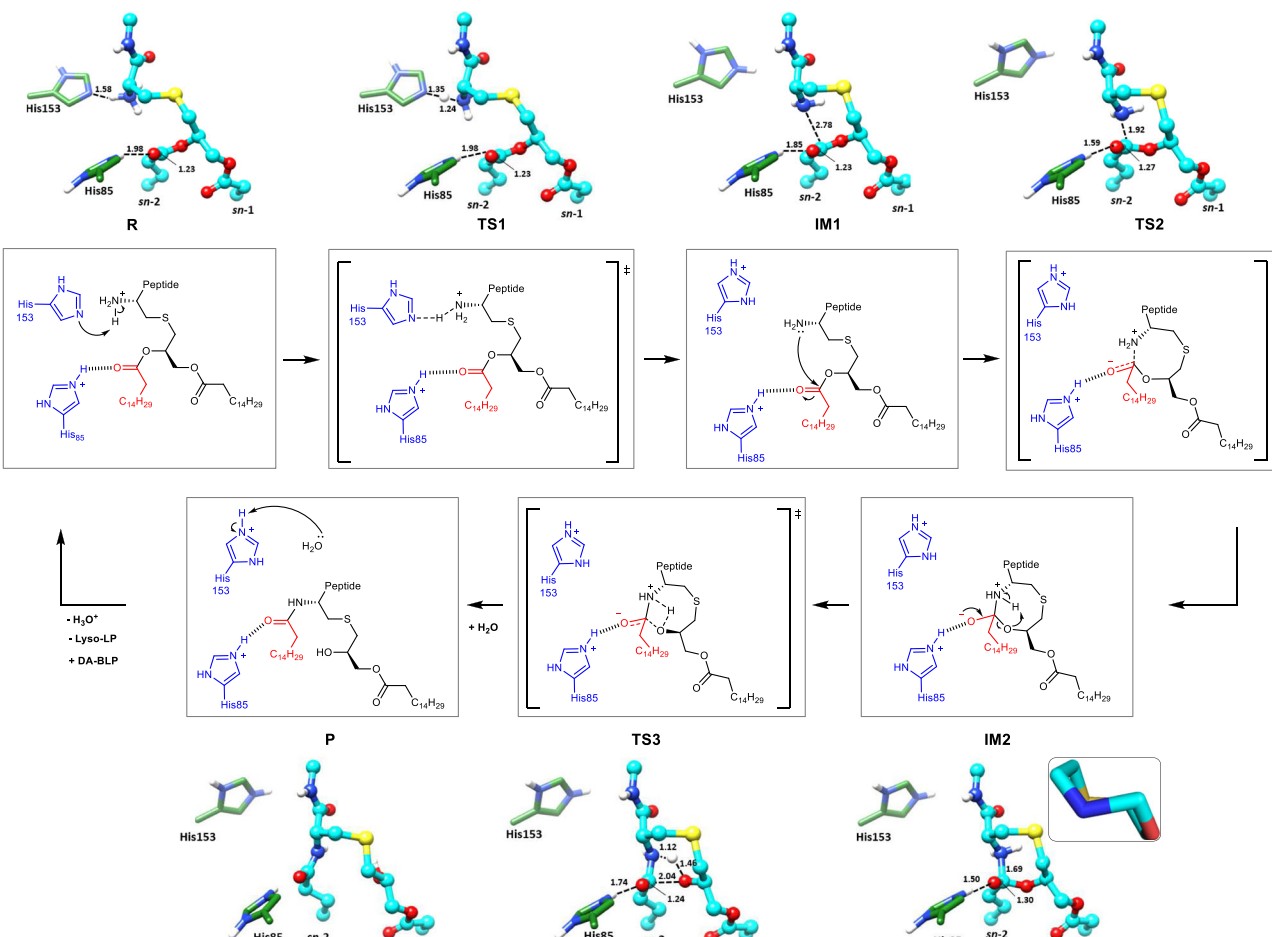

**Fig. 4 Proposed mechanism of the Lit intramolecular transacylation reaction.** (Inner panels). The general acid–base mechanism invokes His153 as the nucleophile which abstracts a proton from the free α-ammonium group of cysteine in the DA-BLP substrate. The newly formed amine (intermediate 1, IM1), attacks the carbonyl carbon at the *sn*-2 position of the DAG forming an eight-membered heterocyclic intermediate with tetrahedral carbon and nitrogen atoms (intermediate 2, IM2). The negative charge on the oxygen bonded to the tetrahedral carbon is stabilised by the proximal His85 which remains protonated throughout the reaction. Collapse of the intermediate gives rise to the lyso-BLP product (P). Deprotonation of His153 by equilibration with bulk water restores the enzyme to its original form in preparation for another round of reaction. Transition state structures (TS1, TS2 and TS3) are shown in square brackets. TS3 includes a transient four-membered ring containing carbon, nitrogen, hydrogen and oxygen atoms that is stabilised presumably by local bonded and non-bonded interactions. (Outer panels) 3D structures of complexes between Lit and reactant, transition states, intermediates and product derived from QM/MM. The eight-membered heterocyclic intermediate (IM2) adopts a chair-like conformation (inset), which was maintained throughout the MD simulations. The acylated cysteine is shown with cyan carbon atoms. The catalytic histidines have green carbons. Polar and hydrogen bond interactions are shown as dashed lines with distances in ångströms. Dashed lines in the transition states represent forming and breaking partial bonds. The bond between the carbon and the non-bridging oxygen in the *sn*-2 acyl chain is identified by a black line.

hydrate undoubtedly contributes to the driving force for the product to depart the binding pocket. The last step in the reaction is the deprotonation of His153 which occurs by facile protonic equilibration with bulk water (Supplementary Fig. 15b). This is accompanied by the return of the imidazole side chain of His153 to face into the binding pocket restoring the enzyme to its original form for another round of reaction (Supplementary Movie 7).

The energy cost associated with this reaction pathway was 21.7 kcal/mol (Supplementary Fig. 15b), which means that the direct mechanism is thermodynamically feasible[37]. It corresponds to a rate constant of $0.19\,min^{-1}$ which agrees well with the experimental apparent turnover number of $0.25\,min^{-1}$. The concurrence lends credence to the proposed general acid–base mechanism.

## Discussion

The work presented here was performed using heterologously expressed and detergent solubilized protein. The structural information was obtained with the enzyme confined to a crystal lattice. By contrast, related studies of Lit by Kurokawa et al.[24], and by the Meredith group[1,22] were carried out, for the most part, with whole cells. Satisfyingly, the crystal structure matches and rationalises many of the observations made with the protein in its more natural in vivo setting. These include the membrane topology of the protein with its four transmembrane helices, intracellular N- and C-termini, and extracellular loops in which reside the catalytic histidines. Indeed, the rationalisation extends to the difficulty in the whole cell work of localising the domains in which the catalytic residues are found. This makes sense because EGD2, which includes His153, is flexible with amphiphilic H4 sitting at the membrane interface. This flexibility is corroborated by the high crystallographic B-factors in this part of the protein and by MDS analysis (Supplementary Fig. 3 and Supplementary Movie 2). Further, alanine-scanning mutagenesis applied to E. faecalis Lit alleles in E. coli, with cell growth rescue as readout, identified a set of residues that map remarkably well to those in the substrate-binding pocket and active site identified on the basis of the crystal structures[1]. This agreement between whole cell and pure protein work lends credence to the physiological relevance of the current findings.

It is worth considering why it is necessary in certain organisms to convert DA-BLP to lyso-BLP and by extension, why Lit is necessary? Another way to pose the question is, What is it that a lyso-BLP does that its DA-BLP counterpart cannot? The question is particularly relevant in the case of Gram-positive bacteria which lack an outer membrane. Such organisms do not have a requirement for Lnt and TA-BLP formation for trafficking via the Lol system as applies in the case of Gram-negative bacteria. Given the compositional make-up of the Lit substrate and product, it cannot be the identity of the fatty acyl chains that distinguishes them because the Lit reaction simply moves one of the chains on the DAG to the amino group of the N-terminal cysteine. Perhaps however the shift alters the relative orientation, depth of penetration, partitioning and/or mobility of the substrate and product in the membrane (Supplementary Fig. 36). These differences may, in turn, influence the activity of the lipoprotein. The effect could be to toggle activity between on and off states or to more gradually modulate the lipoprotein's activity level. An argument against this being a reversible switch must reference thermodynamics. The amide linkage in the product, lyso-BLP, is considerably more stable than the ester in the substrate, DA-BLP. This is borne out experimentally in that wild-type E. faecalis cells were found to have little detectable DA-BLP; the vast majority of the cell's BLPs existed in the more stable lyso-BLP form[22]. Thus,

once formed lyso-BLP does not revert to a DA-BLP. And any change in activity as a result of transacylation is irreversible.

For whatever reason Lit came to exist, it has the advantage over Lnt and the recently discovered LnsAB[38] and AatD[39,40], the only other known amino terminus acylating enzymes, that upon reaction it does not produce a lyso-phospholipid (lyso-PL) as a second product[1]. Lyso-PLs, as the name implies, are potent surfactants that, at high enough concentrations, will effectively punch holes in and eventually solubilise membranes. A million copies of Braun's lipoprotein reside in the cell wall of E. coli, all processed through Lnt[2]. Simultaneous with lipoprotein synthesis, an equal number of lyso-PLs is produced. A typical cell has an estimated 20 million PLs in each leaflet of the inner membrane. The newly formed lyso-PL will therefore appear in the outer leaflet of the inner membrane at high (5 mol%) concentration. If nothing is done to remove this powerful detergent, the integrity of the membrane would be compromised with fatal consequences for the cell. Fortunately, repair systems are in place that shunt lyso-PLs back into benign diacyl-PLs[41,42]. No such repair system exists in Gram-positive bacteria which means that an Lnt-type activity, with lyso-PL as a product, could not be tolerated. It makes sense therefore that Gram-positive organisms, with a need for N-acylated lipoproteins, should have evolved Lit activity. Because the reaction catalysed by Lit is intramolecular, additional substrates and products are not involved, making for an efficient route to N-acylated lipoproteins.

Copper is an essential trace element. Beyond a certain concentration however this transition metal becomes toxic[43]. Copper is approved for public health use as an antibacterial agent especially for sterilising touch surfaces. Its application as a growth-promoter in the diet of food-producing animals has raised concerns for possibly contributing to the rise in antimicrobial resistance[44]. In host/pathogen relationships, the toxic properties of copper have been exploited. Infected macrophages, for example, store copper in phagosomes to combat Staphylococcus aureus. In turn, bacteria have evolved defence mechanisms that employ copper transporters, chaperones and oxidases. These copper-detoxification features are typically found in mobile genetic elements as plasmids or islands in chromosomal DNA. They contribute to the facile development of strains resistant to copper and to elimination by the host immune system. lit2 is part of a copper-sensitive operon. Its expression, upon exposure to copper, coincides with the upregulation of copper-resistance genes[23]. As noted, Lit2 expression will lead to the conversion of DA-BLP to lyso-BLP. In so doing, a potent agonist of the innate defence system's TLRs is converted to the much more benign lyso-BLP. This immunity attenuating response, presumably works in concert with the suite of newly expressed copper-resistance genes to enable the invading organism to fend off attack by the host, and to survive and multiply. Recently, CopL, a lipoprotein in S. aureus, has been shown to bind copper ions with attomolar affinity[45]. CopL has been speculated to play a role in resisting copper killing and in binding excess copper pumped from the cell by efflux proteins. DA-BLPs may function likewise to sequester piped copper. Copper binding proteins may also act as a reserve in times of copper deficiency.

In the absence of experimental proof that Lit transfers the sn-1 and not the sn-1 chain from the DAG to the free amino group of DA-BLP, the same conclusion could perhaps have been reached based on a simple inspection of the unimolecular transacylation reaction. The reaction passes through a heterocyclic intermediate. If it is the chain at sn-1 that gets transferred, an intermediate with a nine-membered ring forms. By contrast, it is an eight-membered ring that forms when transfer is from the sn-2 position (Fig. 4). Different sized rings are characterised by different strain energies[46]. That for a nine-membered ring, at least in the

case of cycloalkanes, is strained to the extent of 2–3 kcal/mol more than its eight-membered counterpart. For this reason alone, and considering the behaviour of carbocycles in particular, it was predictable that the transfer catalysed by Lit involved the *sn*-2 and not the *sn*-1 as the acyl chain donor site. Relatedly, the cyclooctanoid is a motif common to several therapeutically important natural products and is noted for its conformational flexibility. The boat-chair conformation is the most stable. While total synthesis of compounds that include eight-membered rings is experimentally challenging, Lit would appear to have evolved a way of doing so facilely in a reaction vessel some cubic nanometres in volume that is the active site pocket of the enzyme. Indeed, lessons in total synthesis might usefully be gained by inspecting the form of and the chemical constellation that constitutes this highly evolved nanospace.

By two orders of magnitude, lyso-BLPs are less potent TLR agonists than DA-BLPs or TA-BLPs[23]. Thus, Lit may have evolved to enable organisms evade detection and clearing by the host by converting DA-BLP to the more benign lyso-BLP. This would enable bacteria, endowed with Lit, to gain a foothold by stealth by not hyper-activating the innate immune alert system. Thus, Lit may be viewed as a virulence factor and a target for antibiotic development[47]. Resistance pressure and the likelihood of resistance development are much reduced in the case of a non-essential functionality as with Lit. A therapeutic that blocked Lit activity would therefore cause DA-BLPs to accumulate which, in turn, would activate the immune systems to clear the infection.

Based on assays performed with nonimmune HEK-Blue-TLR2/1/6 reporter cells using synthetic and native BLPs, lyso-BLPs have been shown to interact preferentially with the heterodimeric TLR2/TLR6 as opposed to the TLR2/TLR1 receptor pair[23]. This finding was unexpected in light of earlier crystal structure and functional studies showing DA-BLP and TA-BLP binding to and signalling at the TLR2/6 and TLR2/1 receptor systems, respectively[48,49]. Thus, the DAG chains in the two BLPs were accommodated in the large binding pocket of TLR2 while the N-acyl chain of TA-BLP was located in the smaller binding pocket of TLR1. TLR6 remained empty in its complex with TLR2 and DA-BLP. However, from our crystal structure and modelling studies, the two chains in lyso-BLP can come into close contact adopting a side-by-side conformation as might be expected for adjacent chains in a DAG. This would explain how the two chains in the lyso-BLP can bind with TLR2. It may be that the interactions in this compact TLR2/TLR6 complex are energetically more favourable than might apply with the two chains spayed apart, one in TLR1 and the other bound to a half empty TLR2, in the TLR2/TLR1 pair. Regardless of the strength of the interaction, the potency of lyso-BLP TLR signalling is reduced by two orders of magnitude compared to the DA- and TA-BLPs.

## Methods

**Cloning and site-directed mutagenesis**. The wild-type *lit* gene from *B. cereus* strain ATCC 14579 (Uniprot ID: Q813T3_BACCR) was codon optimised for expression in *E. coli* cells and commercially synthesised (GenScript, USA). The synthetic gene was amplified by polymerase chain reaction (PCR) using amplification primers listed in Supplementary Table 6. The resulting PCR product was digested by the restriction enzymes NdeI and XhoI, and ligated into the NdeI/XhoI digested pET28MHL plasmid (GenBank accession number EF456735). The resultant pET28MHL-*lit* plasmid encodes for *B. cereus* Lit protein carrying a M-H$_6$-SSGRENLYFQGH N-terminal extension.

Lit mutants were obtained by site-directed mutagenesis using the Q5® Site-Directed Mutagenesis Kit (New England Biolabs, UK), the wild-type pET28MHL-*lit* construct as template, and specific primers listed in Supplementary Table 6. All constructs were verified by sequencing (Eurofins Genomics, Germany).

**Expression and purification of Lit**. *E. coli* C43(DE3) cells (Lucigen) carrying the pET28MHL-*lit* plasmid were grown in Terrific Broth medium supplemented with 50 μg/mL kanamycin at 37 °C and 180 revolutions per minute (rpm) (Infors HT

Multitron incubator shaker) to an optical density at 600 nm (OD$_{600}$) of 0.5–0.7. At this point, the temperature was reduced to 20 °C, and the gene expression was induced with 0.6 mM isopropyl β-D-1-thiogalactopyranoside (IPTG). After overnight growth for 16–18 h at 20 °C, cells were harvested by centrifugation at 6000×g for 10 min at 4 °C (rotor F10S-6x500y), and the resulting pellets were resuspended in 0.3 g/mL of Buffer A (25 mM HEPES/NaOH, 200 mM NaCl, 10%(v/v) glycerol, 5 mM MgCl$_2$, 2%(w/v) DNASe, final pH 7.5). Cells were disrupted by four passages at 20,000 p.s.i and 4 °C through a high-pressure homogeniser (Emulsiflex-C5, Avestin®). Cell debris was removed by centrifugation at 30,000 × g for 30 min at 4 °C and the supernatant was ultracentrifuged at 150,000 × g for 90 min at 4 °C (rotor Ti-45). The membrane pellet was resuspended in 10 mL of Buffer B (25 mM HEPES/NaOH, 200 mM NaCl, 10%(v/v) glycerol, 1.5%(w/v) lauryl maltose neopentyl glycol (LMNG), final pH 7.5) per gram of wet membranes and solubilised for 16 h at 4 °C on a Nutator mixer (VWR). Insoluble material was removed by ultracentrifugation at 150,000×g for 30 min at 4 °C. The supernatant was supplemented with 10 mM imidazole and mixed with 3 mL of TALON® Metal Affinity Resin (Clontech Laboratories, Inc.) pre-equilibrated with Buffer C (25 mM HEPES/NaOH, 200 mM NaCl, 10%(v/v) glycerol, 0.02%(w/v) LMNG, final pH 7.5). After incubation for 4 h at 4 °C, the suspension was poured into an empty 1.5 × 12 cm Econo-Pac® chromatography column (Bio-Rad). The resin was washed with 25 mL of Buffer D (25 mM HEPES/NaOH, 200 mM NaCl, 10%(v/v) glycerol, 0.02%(w/v) LMNG, 20 mM imidazole, final pH 7.5), 25 mL of high-salt Buffer E (25 mM HEPES/NaOH, 500 mM NaCl, 10%(v/v) glycerol, 0.02%(w/v) LMNG, 20 mM imidazole, final pH 7.5) and 25 mL of Buffer D. After additional washes with Buffer D containing increasing concentrations of imidazole (10 mL at 40, 60 and 80 mM), the protein was eluted in 6 mL of 1 M imidazole in Buffer D. The eluted protein was further purified by size exclusion chromatography (SEC) using a HiLoad 16/60 Superdex 200 column (GE Healthcare) pre-equilibrated with Buffer F (25 mM HEPES/NaOH, 200 mM NaCl, 10%(v/v) glycerol, 0.02%(w/v) LMNG, final pH 7.5). The purified protein was concentrated to ≥12 mg/mL with a 50 kDa molecular weight cut-off (MWCO) concentrator (Amicon® Ultra, Millipore). Aliquots of the purified protein were flash-frozen in liquid nitrogen and stored at −70 °C.

Expression and purification of Lit wild-type and mutant proteins for enzyme assays were performed as described above except that the buffer used for SEC was Buffer G (25 mM Tris/HCl, 80 mM NaCl, 10%(v/v) glycerol, 0.02%(w/v) LMNG, final pH 8.3).

For selenomethionine (SeMet) labelling, cells were grown in M9 minimal medium to an OD$_{600}$ of 0.6. Methionine biosynthesis was suppressed[50] by adding 100 mg/L of Lys, Phe and Thr, 50 mg/L of Ile, Leu and Val and 60 mg/L of L-SeMet. After 15 min, *lit* gene expression was induced with 0.4 mM IPTG for 18 h at 20 °C.

Purification of SeMet-labelled Lit was performed as described above for the native, wild-type protein with the following modifications. The cell lysis and membrane solubilisation buffers were supplemented with 2 mM DTT, 1 mM PMSF and a tablet of cOmplete™ EDTA-free protease inhibitor cocktail per 50 mL of buffer. The detergent extract was supplemented with 10 mM imidazole and mixed with 5 mL of nickel-charged resin (Ni-NTA agarose, Qiagen) pre-equilibrated with Buffer H (25 mM HEPES/NaOH, 200 mM NaCl, 10%(v/v) glycerol, 2 mM DTT, 1 mM PMSF 0.05%(w/v) LMNG, cOmplete™ EDTA-free protease inhibitor cocktail (1 tablet/50 mL), final pH 7.2). After incubation at 4 °C for 4 h, the resin was washed twice with 50 mL of Buffer H containing 20 and 30 mM imidazole. The protein was eluted with 30 mL of Buffer H containing 600 mM imidazole and concentrated to 10 mL using a 50 kDa MWCO concentrator. The buffer was exchanged to Buffer I (25 mM HEPES/NaOH, 200 mM NaCl, 10%(v/v) glycerol, 5 mM β-ME, 0.05%(w/v) LMNG, final pH 7.2) using PD10 desalting columns (GE healthcare). A second affinity chromatography purification was performed by supplementing the protein with 10 mM imidazole and incubating it with 3 mL of TALON Metal Affinity Resin for 12 h at 4 °C. The resin was washed with 25 mL of high-salt Buffer J (25 mM HEPES/NaOH, 500 mM NaCl, 10%(v/v) glycerol, 5 mM β-ME, 0.05%(w/v) LMNG, 20 mM imidazole, final pH 7.2) and 10 mL Buffer I containing 40 mM imidazole. The protein was eluted in 3 × 10 mL of Buffer I containing 500 mM imidazole. The elution fractions were pooled, concentrated to 1.8 mL and subjected to SEC on a HiLoad 16/60 Superdex 200 column equilibrated with 25 mM HEPES/NaOH, 200 mM NaCl, 10%(v/v) glycerol, 0.05%(w/v) LMNG, 2 mM TCEP, final pH 7.2. The purified SeMet-labelled protein was concentrated to 15.6 mg/mL for use in crystallisation trials.

**Protein quantitation**. The concentration of Lit wild-type and mutants proteins was determined with a Nanodrop-1000 spectrophotometer (Nanodrop Technologies, Wilmington, DE, USA) using the molecular weights and extinction coefficients at 280 nm calculated with the ExPASy ProtParam tool[51].

**Peptide synthesis**. Lit substrates FP2, (*S*)-DAG-FP2 and dFP2 were prepared by manual 9*H*-fluoren-9-ylmethoxycarbonyl/*tert*-butyl (Fmoc/*t*Bu) solid phase peptide synthesis (SPPS) on Rink amide resin. The initial resin was Fmoc-deprotected using 20%(v/v) piperidine/N,N-dimethylformamide (DMF) for 2 × 10 min. Loading of the first amino acid was performed with Fmoc-Lys(N$^ε$-4-methyltrityl)-OH (4 equiv), benzotriazol-1-yl-oxytripyrrolidinophosphonium hexafluorophosphate (PyBOP) (4 equiv) and N-methylmorpholine (NMM) (8 equiv) in DMF for 45 min. On-resin introduction of the fluorophore, Abz, to the lysine side chain was achieved via 4-methyltrityl deprotection with 5%(v/v) trifluoroacetic acid (TFA),

5%(v/v) triethylsilane (TES) and 90%(v/v) $CH_2Cl_2$ for $5 \times 1$ min followed by coupling of Boc-Abz-OH (3 equiv) with PyBOP (3 equiv) and NMM (6 equiv) in DMF for 45 min. To complete the assembly of each peptide on-resin, iterative cycles of Fmoc deprotection using 20%(v/v) piperidine/DMF for $2 \times 10$ min and coupling of Fmoc-amino acid (3 equiv) with PyBOP (3 equiv) and NMM (6 equiv) in DMF for 45 min were carried out. Peptides were subsequently cleaved from the resin and fully deprotected upon treatment with 95%(v/v) TFA, 2.5%(v/v) TES and 2.5%(v/v) $H_2O$ for 90 min. Final peptides were precipitated in $Et_2O$ and purified by normal phase silica gel column chromatography. Further synthetic details with compound characterisation are provided in the Supporting Information.

Modified cysteine building blocks, Fmoc-Cys((S)-2,3-bis(palmitoyloxy)propyl)-OH and Fmoc-Cys((R)-2-((hexadecanoyl-$d_{31}$)oxy)-3-(palmitoyloxy)propyl)-OH were synthesised as described in the Supporting Information.

**TLC-based activity assay**. The activity of Lit was assayed in 50 µL of reaction mixture containing 150 µM FP2 in assay buffer (25 mM MES/NaOH pH 5.4, 200 mM NaCl, 0.02%(w/v) LMNG). The lipopeptide was added from a stock solution at 10 mg/mL (9.1 mM) in dimethyl sulfoxide (DMSO). The reaction was initiated by adding 10 µM Lit and was carried out at 37 °C without shaking. The reactions were stopped after the indicated time by flash freezing in liquid nitrogen.

To aid extraction of the lipophilic peptide substrate and product from the reaction mix, 50 µL of 70%(v/v) ethanol was added to the frozen reaction mix samples, followed by vortexing for 10–20 s at 20 °C until an homogenous emulsion was obtained. Lipophilic substrate and product were extracted by vortexing 30 µL of chloroform with the emulsion for 60 s. Phase separation was facilitated by centrifugation in a benchtop centrifuge for 2 min at 13,000×g and 20 °C, and the lower chloroform phase was transferred into a 1.5-mL Eppendorf tube. The tube was left open in a fume hood for 10 min at 20 °C to passively evaporate excess chloroform. The tube was centrifuged for 2 min at 13,000×g and 20 °C, and all of the collected organic phase was spotted on a silica gel 60 $F_{254}$ TLC plate (Sigma Aldrich). The plate was placed in a desiccator at 20 °C under high vacuum (50 mbar) for 10 min to remove residual DMSO. Chromatography was carried out with a mobile phase consisting of $CHCl_3$:MeOH:28% $NH_4OH_{aq}$ (8:2:0.1 by vol.) and the substrate and the product were directly visualised on the TLC plate under UV light at 254 nm through quenching of the fluorescence indicator in the TLC plate by the Abz moiety. A digital image of the UV-illuminated TLC plate was taken using a smartphone, and the substrate and product band intensities were quantified by image analysis using ImageJ[52]. The results were plotted using Prism 8 (GraphPad).

**Lyso-FP2—NMR analysis**. NMR spectroscopy was used to confirm the identity of the Lit product. Several Lit reactions were set using a total of 100 and 500 µg of protiated (FP2) or deuterated (dFP2) substrate, respectively, to generate enough material for NMR analysis. The reactions were incubated at 37 °C for 18 h and stopped by flash-freezing in liquid nitrogen. The substrate and product were extracted as described above and separated on a preparative silica gel TLC plate (Sigma Aldrich). The product was recovered by scraping the silica from the plate with a clean spatula and washing it twice with $CHCl_3$:MeOH (19:1, v/v). The washings were combined and the silica pelleted by centrifugation 1000×g for 2 min. The supernatant was collected and the solvent was evaporated under a stream of nitrogen. The product was thoroughly dried in vacuo. Following preparative TLC, each isolated product (lyso-FP2 and lyso-dFP2) was dissolved in DMSO-$d_6$ (0.6 mL). NMR spectra were recorded on a 600 MHz Bruker Avance II spectrometer equipped with a 5 mm TCI CryoProbe operating at a frequency of 600.13 MHz ($^1H$ NMR) or 150.6 MHz ($^{13}C$ NMR) at 25 °C. HSQC and TOCSY NMR experiments were carried out to aid in structure elucidation (Supplementary Figs. 23–34). NMR data was analysed using MestReNova v6.0.2–5475.

**Copper–dFP2 interaction—NMR analysis**. A solution of $CuSO_4.5H_2O$ (0.023 mg, 0.0915 mol) in DMSO-$d_6$ (60 µL) was added to a solution of dFP2 (1.00 mg, 0.915 µmol) in DMSO-$d_6$ (600 µL). The sample was mixed, and NMR data was collected and analysed as described above for the lyso-FP2 product.

**Crystallisation**. Crystals of wild-type native and SeMet-substituted Lit were obtained by mixing the protein at 12–16 mg/mL with monoolein (NuChek) in a 2:3 protein:lipid volume ratio using the twin-syringe mixing method[26,53]. Aliquots of 30 nL of protein-laden mesophase were spotted on 96-well glass sandwich plates and overlain with 800 nL of precipitant solution using a Gryphon LCP (Art Robbins Instruments) or a Xantus (SIAS) crystallisation robot[54]. The plates were stored at 20 °C in a crystallisation incubator (RUMED® 3101, Rubarth Apparate GmbH, Germany) or in an incubator/imager (Rock Imager® 1500, Formulatrix, USA) for crystal growth. Crystals typically appeared within 24 h and grew to full size in 2 weeks. Monoclinic P$2_1$ crystals grew in a precipitant solution composed of 100 mM sodium citrate/HCl, pH 5.6, 40%(v/v) polyethylene glycol (PEG) 400, 100 mM ammonium sulfate and 200 mM sodium formate. The plate-shaped crystals measured up to 400 µm in length. Orthorhombic P$2_12_12$ crystals, rod-shaped and measuring $10 \times 10 \times 50$ µm$^3$, grew in a precipitant solution composed of 100 mM sodium citrate/HCl, pH 5.5, 75–150 mM NaCl and 36–44%(v/v) PEG 200.

For co-crystallisation trials with the substrate (R)-DAG-FP2, purified Lit H85A and H85R mutant proteins were diluted to 2 mg/mL in Buffer G, mixed with a fivefold molar excess of (R)-DAG-FP2 in DMSO, and incubated for 16–20 h at 4 °C on a Nutator mixer (VWR) or a rotator (SB3, Stuart®, UK) circulating at 8 rpm.

For the H85A mutant, the cubicon method[33] was used to concentrate the protein-peptide mixture in the cubic mesophase. After 5 to 7 rounds of reconstitution, the equivalent starting protein concentration was 12–14 mg/mL. For the H85R mutant, the protein-peptide solution was concentrated to 15 mg/mL with a 0.5-mL concentrator (50 kDa MWCO, Amicon® Ultra, Millipore) and mixed with monoolein in a 2:3 protein:lipid volume ratio. LCP crystallisation plates were set as described above for the wild-type protein. Crystals of the H85A mutant were obtained in a precipitant composed of 100 mM sodium citrate/HCl, pH 5.5, 75–150 mM NaCl and 36–44%(v/v) PEG 200. In contrast, crystals of the H85R mutant grew in a precipitant composed of 100 mM MES/NaOH, pH 6.0, 125–200 mM lithium citrate tribasic tetrahydrate and 36–44%(v/v) PEG 400. For both mutants, monoclinic C2 crystals appear within 24 h and grew to a maximum dimension of $20 \times 60$ µm$^2$ in 2 weeks.

Crystals were harvested from the lipid cubic phase using Dual Thickness MicroLoops LD or MicroMounts loops (MiTeGen), and snap-cooled and stored in liquid nitrogen without added cryo-protectant. The in meso in situ macromolecular X-ray crystallography (IMISX) method was used to grow and to collect data on SeMet-labelled Lit crystals[55].

**Diffraction data collection and processing**. X-ray diffraction experiments were carried out at 100 K on protein crystallography beamlines X06SA-PXI at the Swiss Light Source, Villigen, Switzerland, and I24 at the Diamond Light Source, Didcot, UK. Measurements were made in steps of 0.1–0.2° at speeds of 1–2° per s with either the EIGER 16 M or the PILATUS 6M-F detector operated in a continuous/shutterless data collection mode. For SeMet SAD phasing, diffraction data were collected on SeMet-derivative B. cereus Lit (Lit-Se) crystals using both rotation and serial crystallography methods[55] at wavelength and flux values of 0.97882 Å and $10^{12}$ photons per s, respectively. Native data from wild-type Lit (Lit_1, Lit_2) and mutant form (LitH85A, LitH85R) crystals were all measured using the rotation method at wavelengths and flux values of 0.9686 Å and $4.1 \times -10^{12}$ photons per s, respectively. The data set for Lit-Se crystals was measured with a $20 \times 10$ µm$^2$ X-ray beam size at beamline X06SA-PXI. Native data sets from Lit_1, Lit_2, LitH85A and LitH85R crystals were measured with a $20 \times 20$ µm$^2$ X-ray beam size at beamline I24 DLS. Data were processed with XDS[56] for Lit-Se and LitH85R with a resolution cut-off at I/σ(I) of 1. For Lit_1, Lit_2 and Lit_H85A, the data sets were processed using autoPROC/STARANISO[57,58] with the resolution cut-offs applied based on ellipsoidal diffraction limits. All of the data were scaled and merged with XSCALE[56]. A SeMet derivative data set to 3.2 Å was obtained by merging the data from 16 single crystals with 360° per crystals using the data acquisition software suites (DA+)[59] and 143 small wedge data of IMISX crystals with 10–20° per crystals using an automated collection programme[60] available at X06SA-PXI. Data sets for Lit_1, Lit_2, LitH85A and LitH85R were collected from single crystals with 360° wedges to 2.27, 1.95, 2.43 and 2.20 Å resolution, respectively. Data collection parameters are summarised in Supplementary Table 1.

**Structure solution and refinement**. The SAD method was employed for structure solution using data sets from Lit-Se crystals. A substructure with $CC_{all/weak}$ values of 44.36/16.51 was obtained with 10,000 SHELXD trials using the HKL2MAP[61] interface of SHELXC/D/E. Additional phase improvement and iterative auto-model building were carried out with CRANK2[62] using the substructure from SHELXD, the Lit sequence and a high-resolution native data set obtained with Lit_1 crystals. The Lit model with two molecules in the asymmetric unit was fully built automatically by CRANK2 and was manually adjusted using Coot[63]. The structures of Lit_2, LitH85A and LitH85R were obtained using Phaser[64] with chain A of Lit_1 as the searching template. Phenix.refine[65] and BUSTER[66] were used during the refinement of all structures. Phasing and refinement statistics are reported in Supplementary Table 1. Figures of molecular structures were generated with PyMOL[67].

**Sequence conservation analysis**. Sequences homologous to Lit were retrieved from the UniProt Reference Proteomes database[68] using the HmmerWeb server[69,70]. The E value cut-off was set to 0.0001, and other parameters were kept to their default values. The search converged after six iterations, and 788 sequences were retrieved and aligned with Clustal Omega[71]. Conservation scores were calculated using the Consurf server[72,73] and mapped on the Lit crystal structure using Pymol[67]. Sequence logo was produced using the WebLogo3 server[74]. A phylogenetic tree was constructed in MEGA X[75] using the neighbour-joining method[76].

**MD and QM/MM simulations**
*System preparations for MD simulations*. The crystal structure of Lit (MolB) at 1.95 Å resolution was used as an initial structure for MD simulations. The MD substrate, Cys-Ser-Ser-Lys, with the cysteine residue attached to a dipalmitoylglyceryl (R-stereoisomer) group was built using Maestro 2018–4 in the Schrodinger software suite[77]. The structure of Lit with two monoolein molecules in the vicinity of catalytic His85 and His153 provided a good starting point for substrate docking. After removing the structured monooleins and water molecules from the Lit structure, the MD substrate was placed into the active site with the sn-2 chain

toward the catalytic histidines. The substrate-Lit complex was then subjected to a short energy minimisation using Maestro 2018-4[77]. The product-Lit complex was generated by removing the *sn*-2 chain from the substrate DAG and linking it via an amide bond to the α-amino group of cysteine. The product complex was also energy optimised in Maestro 2018-4[77]. Similarly, the Lit complex with lipopeptide monoacylated at the *sn*-1 position was created from the substrate-Lit complex. The structures of intermediate 1 (IM1) and intermediate 2 (IM2) complexes were obtained from the QM/MM study. The force field parameters for the substrate, product, monoacylated lipopeptide, IM1 and IM2 were developed in the Antechamber programme of AmberTools18[78] using the GAFF force field[79].

The protonation states of titratable residues at pH 7.0 were predicted using the PROPKA programme[80] available in Maestro 2018-4[77]. Aspartate, glutamate, arginine and lysine residues were assigned a negative or a positive charge, as appropriate. Non-catalytic histidine residues were modelled as neutral, with a hydrogen atom bound to either the delta or epsilon nitrogen depending on which tautomeric state optimised the local hydrogen-bonding network. Protonation states of catalytic His85 and His153 were assigned taking into consideration their local environment, QM/MM, and mutagenesis data. The side chain of His153 was in the neutral form with a proton at the delta nitrogen in the MD simulations of Lit bound to substrate and to monoacylated lipopeptide, and in the empty form. His153 was in the charged imidazolium form in MD simulations of Lit bound to the product and in the IM1 and IM2 states. Initial exploratory simulations were performed with His85 in positively charged and neutral forms. His85 in the charged form was chosen for the final MD simulations because the QM/MM analysis showed that the eight-membered heterocyclic intermediate (IM2) of the transacytlation reaction can only form in the presence of protonated His85.

Six Lit biosystems were used in this study. They include the empty form (where the two structured monoolein molecules have been removed from the active site pocket), substrate-, product- and monoacylated lipopeptide-bound forms, and IM1 and IM2 states in a hydrated lipid bilayer. These were constructed to study enzyme dynamics and the reaction mechanism. The lipid bilayer consisted of 1-palmitoyl-2-oleoyl-*sn*-glycero-3-phosphocholine (POPC) and 1-palmitoyl-2-oleoyl-*sn*-glycero-3-phosphoglycerol (POPG) in a 3:1 mole ratio. The position of the Lit molecule across the lipid bilayer was established using the Orientation of Protein in Membranes (OPM) server[81]. The CHARMM-GUI Lipid Builder[82] was used to insert the Lit structure into the lipid bilayer consisting of 120 lipid molecules in each leaflet of the membrane. TIP3P water molecules[83] were used to solvate the bilayer and counterions were added using Monte Carlo simulations at a concentration of 0.15 M NaCl. The final systems comprised ~88,000 atoms with a box dimension of $95 \times 95 \times 100$ Å$^3$.

*MD simulations protocols*. All MD simulations were performed using the Compute Unified Device Architecture (CUDA) version of particle-mesh Ewald molecular dynamics (PMEMD) in Amber18[78,84,85] on graphics processing units (GPUs). All systems were subjected to energy minimisation using the steepest descent (1000 steps) followed by the conjugate gradient (1000 steps) methods. Firstly, the protein and bilayer were restrained using a potential of 5 kcal mol$^{-1}$ Å$^2$ and only solvent and ions were allowed to relax. This was followed by a full minimisation of the entire systems using steepest descent (5000 steps) and conjugate gradient (5000 steps) methods.

The systems were heated for 100 ps from 0 to 100 K in the NVT ensemble using a Langevin thermostat with harmonic restraints of 5 kcal mol$^{-1}$ Å$^{-2}$ on the non-hydrogen atoms of the lipids, protein and lipopeptide. Initial velocities were sampled from a Boltzmann distribution. Next, the system was heated to 310 K over 300 ps in the NPT ensemble. Equilibration was performed at 310 K and 1 bar in an NPT ensemble for 100 ns in five steps: (I) the membrane lipid head groups were allowed to move, while the whole system was restrained using 5 kcal mol$^{-1}$ Å$^{-2}$ for 5 ns; (II) the lipid bilayer was allowed to move, while the rest of the system was restrained using 5 kcal mol$^{-1}$ Å$^{-2}$ for 10 ns; (III) the lipid bilayer including water and ions were allowed to move, while the protein was restrained using 1 kcal mol$^{-1}$ Å$^{-2}$ for 15 ns; and (IV) the whole system was allowed to equilibrate without any restrains for 70 ns. The final production step of 500 ns was run at 310 K and 1 bar in the NPT ensemble using the Langevin thermostat and Monte Carlo barostat. The simulations were performed using a time step of 2 fs. Non-bonded interactions were cut-off at 10.0 Å and long-range electrostatic interactions were calculated using PMEMD[85,86]. The SHAKE algorithm was used to constrain bond lengths[87]. Five replica runs of 500 ns MD simulation each were performed for all systems. The FF14SB[88] and Lipid14[89] force fields were used in all simulations. The simulations were performed on Kelvin2 and JADE clusters.

*MD simulations trajectory analysis*. Trajectory reimaging, root mean square deviation, root mean square fluctuations, distances, hydrogen bonding, principal component analysis and cluster analysis were conducted using the AmberTools18 CPPTRAJ package[90]. Visual molecular dynamics (VMD) was used for visualisation and movies[91]. PyMOL and Maestro 2018-4 were used to make images. The residue–ligand interaction energy was calculated using the 'namdenergy.tcl' script version 1.6 of NAMD 2.13[92].

*QM/MM adiabatic potential energy calculations*. Structures for the QM/MM calculations were obtained from the cluster analysis of production trajectories of Lit in

complex with the lipopeptide substrate and product. The structures were first subjected to 2000 steps of steepest descent energy minimisation followed by 2000 steps of conjugate gradient minimisation using Amber18[78]. The lipid bilayer was excluded in the QM/MM setup, but the water molecules that were present near the active site from the MD simulations were included. The QM/MM calculations monoacylated was performed using the ChemShell suite 2019[93]. All the quantum mechanical (QM) calculations were performed with the density functional theory (DFT) using the B3LYP functional with def2-SVP def2/J basis set. The D3 dispersion corrections with BJ damping were also included in the DFT calculations[94–96]. The QM calculations was run using the ORCA 4.2.0 package[97] and the MM calculations were computed using the DL_POLY MD simulation package[98]. The FF14SB force fields[88] were used for the MM calculations. The RIJCOSX approximation with the TightSCF criteria, Grid4 and GridX4 were used in the QM calculations. The effect of the protein environment on the polarisation of the QM wave function was described by the electronic embedding scheme[99]. The QM region (74 atoms in total) consists of DAG-Cys, the α-amino group of the cysteine residue and His153 and His85 residues with an overall charge of +2. The residues and water molecules, which were within 8 Å of the acylated cysteine were allowed to move freely and the rest of the system was frozen during the geometry optimisation. Hydrogen link atoms were used to saturate the dangling bonds at the QM region. The reaction coordinate for the formation of the eight-membered cyclic intermediate (IM2) starting from the product structure was defined as a linear combination of the distances d1 and d2 (Supplementary Fig. 16).

The transition sate structures associated with the reaction mechanism of Lit were obtained by running the minimum energy path calculations using the nudged elastic band method[100,101]. The transition state structures were then subjected to full optimisation using the dimer method[102]. The transition state structures were validated by the presence of a single imaginary frequency at 310 K using the thermal keyword in the ChemShell suite[93]. The distort keyword was then used to obtain the corresponding local minima structures connecting the transition states. Rate constants ($k_{cat}$) were calculated using the Arrhenius equation at $T = 310$ K using a pre-exponential factor derived from transition state theory,

$$k_{cat} = \kappa (K_B T / h) e^{-(Ea/RT)} \tag{1}$$

where $\kappa$ is the reflection coefficient that was set equal to 1, $Ea$ is the activation energy barrier, $R$ is the gas constant, $K_B$ is Boltzmann's constant, and $h$ is Planck's constant[103,104].

**Reporting Summary**. Further information on research design is available in the Nature Research Reporting Summary linked to this article.

## Data availability
Atomic coordinates and structure factors have been deposited in the Protein Data Bank (PDB) with accession codes 7B0O (apo Lit wild-type, monoclinic $P2_1$ form), 7B0P (apo Lit wild-type, orthorhombic $P2_12_12$ form), 7B0Q (LitH85A, monoclinic $C2_1$ form) and 7B0R (LitH85R, monoclinic $C2_1$ form). The atomic coordinates of the QM region used in the QM/MM are available as a Supplementary Dataset 1. Other data supporting the findings of this manuscript are available from the corresponding author upon reasonable request. Source data are provided with this paper.

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

## Acknowledgements

We thank L Figur, M Elhantati and J Bond for assistance with protein production and purification, and with crystallisation trials; synchrotron facility scientists at the Diamond Light Source (beamline I24) and the Swiss Light Source (beamlines X06SA and X10SA) for assistance and support, and past and present members of the Membrane Structural and Functional Biology group for assorted contributions to the study. This work was supported in part by Science Foundation Ireland awards 16/IA/4435 (M.C.) and 15/CDA/3310 (E.M.S. and K.B.), an Irish Research Council fellowship GOIPG/2016/1238 (K.B.), a German Research Foundation grant WE 6084/1-1 (D.W.), by the European Union's Horizon 2020 research and innovation programme under the Marie-Skłodowska-Curie grant agreement No. 701647 (C.-Y.H.) and the Northern Ireland Department of Agriculture, Environment and Rural Affairs Award No. 05935-NAGpro (I.G.T.). W.S. acknowledges the support by Research England's Expanding Excellence in England (E3) Fund. This project made use of computational time on Kelvin-2 (grant no. EP/T022175/1); and JADE granted via the UK High-End Computing Consortium for Biomolecular Simulation, HECBioSim (hecbiosim.ac.uk), supported by EPSRC (grant no. EP/R029407/1).

## Author contributions

M.C. developed the overall research plan; S.O. performed cloning and site-directed mutagenesis, produced and purified the proteins, performed in meso crystallisation screening and optimised crystallisation conditions; D.W. developed and optimised the activity assay. D.W. and S.O. performed activity assays. K.B. performed fluorescent substrates synthesis and purification, and collected and analysed NMR data; S.O., D.W., C.-Y.H. and V.O. collected X-ray diffraction data; C.-Y.H. and V.O. solved structures; W.S. and I.G.T. performed MD and QM/MM simulations; all authors analysed data; S.O., E.M.S., I.G.T., V.O. and M.C. supervised the research; M.C. wrote the main manuscript with input from all authors.

## Competing interests

The authors declare no competing interests.
