## [Peer Review File · Nature Communications]

REVIEWER COMMENTS

Reviewer #1 (Remarks to the Author):

This research article presented structure of lipoprotein intramolecular transacylase (Lit) in Gram-positive bacteria and discussed the potential reaction mechanism. The authors presented several crystal structures of the Lit, a novel enzyme in modifying bacterial lipoproteins. The authors also coupled the study with functional assays, chemical biology, mutagenesis, and computational analyses to provide more insight to Lit functions. This was a well-thought out study providing lots of insight into Lit functionality and mode of action, and the authors concluded with proposed mechanisms based upon all the evidence presented. The study should provide an additional tool in understanding bacterial pathogenesis and evasion of host immunity.

The crystal structures provided insight into the possible binding modes of the endogenous ligand by way of the two monoolein (MO) molecules. Unfortunately due to the unreleased nature of the PDB coordinates, it would have been great to be able to see the 2Fo-Fc map around MO molecules in each PDB molecule. The authors showed partial views in Fig.S18. In Fig.S18B the density for FP2 is incomplete, unfortunately. Would it be more beneficial also for the authors to present the 2Fo-Fc maps of the active sites from Lit_2 structure (1.94 Å structure)?

Crystallographic table details for PDB:7B0O (Lit_1) showed that CC1/2 is 0.96 (0.18) - is this a typo?

Line 128: two structures are solved and are they look highly similar? If so, what is the RMSD?

An interesting method that the authors brought up is the use of the cubicon method for concentrating H85A-peptide mutant to eventually set up crystallization trials.

Overall this is a novel enzyme and the authors utilized many different methods to elucidate its mechanism of action. Taken into context within the whole system of lipoprotein modifying enzymes, this study represents a breakthrough in the potential to target bacterial pathogens.

Reviewer #2 (Remarks to the Author):

I have read the manuscript "Structural basis of the membrane intramolecular transacylase reaction responsible for lyso-form lipoprotein synthesis" by Olatunji, S. et al that has been submitted for consideration for publication in Nature Communications. The study reports the crystal structure with biochemical characterization of Lit, an enzyme that remodels lipoproteins. Lipoproteins are ubiquitous components of bacterial membranes that play central roles in both bacterial physiology and infection- as such, the report should appeal to a wide swath of the readership. Lit is also a unique enzyme with respect to activity and structure, as prior insights regarding enzymes catalyzing intramolecular transacylations are rare. Examples of 8-member ring intermediates are equally rare- thermodynamic computation/simulations and in vitro assays provide some support for this intermediates though its existence must remain qualified. Oncemore, this is a timely report given the rapid advancements in the field. Overall the paper is well written, logical to follow, and corroborates/significantly advances prior knowledge regarding this enzyme.

The introduction needs clarifications/corrections as suggested below, some of the speculation regarding specific residues can be trimmed, but otherwise, I am enthusiastic about the manuscript.

Specifics Comments/Suggestions:

1. Abstract, line 23-24: Counting lipoprotein enzyme targets is more complex than implied with this sentence as Meredith/Goetz labs recently reported another N-acylation system in *Staphylococcus aureus* LnsAB *mBio*. 2020 Jul 28;11(4):e01619-20. doi: 10.1128/mBio.01619-20). Also should Lol lipoprotein (LP) transport system be counted here? Please update and/or rework this sentence.
2. line 65- "dagylated"- terminology here and throughout is slang
3. line 66-67- This passage is confusing- all LPs in gram negative (at least model *E coli*) are matured by Lnt to form TA-LP, regardless of whether they are destined for retention in inner or eventually transported to outer membrane. While they must be triacylated for efficient recognition by Lol, the peptide/exclusion signal downstream of Cys is thought to determine further trafficking to OM. Please revise.
4. Fig 1: While overall TLR2/1/6 signaling is indeed extremely weak for lyso-LP, neutralizing TLR1 or TLR6 antibody studies, and TLR2 or 1 specific deletion NF-kB reporter HEK strain assays actually suggested residual lyso-LP induced signaling more likely is from TLR2/6 than TLR2/1 (*J Bacteriol* 2019 Jun 10;201(13):e00195-19. doi: 10.1128/JB.00195-19). Please adjust figure to indicate (i.e. "weak TLR2" signaling for Lit)- NOTE: this is later covered in Discussion section, but figure should still be adjusted to match text
5. line 69- indicate which acyl chain of PE is utilized by Lnt
6. line 82-84: Ref 24, 25 Classifying these as BLPs introduces unnecessary confusion as the (already established) generic "lipoprotein" nomenclature is unfortunate. As noted by authors, the references describe protein acylation on an N-terminal Gly substrate, that are not processed by Lgt, that do not have a diacyl-glycerol modified Cys, no canonical lipobox signal, and are SpsI (not SpsII) dependent. Lumping together with canonical "BLP" as discussed in manuscript is awkward. Consider removing for clarity.
7. line 201-203: Many of these (analogous) residues, including the two His and aromatic residues, have been shown to be functionally important for Lit from *E faecalis* in vivo using an alanine scan library (Ref 1 in paper)- how do those reported here compare?
8. line 382-383/409-410- Mass spec evidence has been provided for which of the two acyl chains is transferred (i.e. sn2) using "mixed chain" DAG (unsaturated Sn2 and saturated Sn1 acyl groups) with isolated membranes in vitro, albeit cyclic transfer reverse acyl-enzyme is unclear
9. Paragraph beginning at Line 503, Lyso-PE: the text here reads quite similar to that which has already been proposed in the discussion of reference 1 and should be cited
10. line 559- "pumped" copper?
11. line 561-577- This is an awkward way to end the paper- I would suggest trimming as the more pertinent aspects can be incorporated earlier in the mechanism part of text where the unique 8-member intermediate is proposed and first discussed

Reviewer #3 (Remarks to the Author):

Dear Editor,

Olatunji and co-workers report the crystal structure of lipoprotein from *Bacillus cereus* and couple it with functional assays, mutagenesis, chemical biology, and computational analysis to establish its mechanism of action. This is a well-conducted study and suitable to be published on the *Nature Communications*.

In general, the study is very interesting; however, I have the following concerns on the manuscript which should be addressed before publication

- 1) The authors should provide in the introduction a critical comparison between all four lipoproteins, highlighting major differences
- 2) A potential energy surface (PES) obtained from QM/MM calculations for the whole studied mechanism should be included on the main text of the manuscript.
- 3) A Table containing relevant geometric parameters (atomic distances) during QM/MM PES should be provide on the main text of the manuscript and related to their respective states.
- 4) As a total of 500 ns of MD was performed to system, I suggest a free energy landscape (FEL) from the principal component analysis (PCA) to avail the most favourable representative (more stable) structure. Finally, a suitable discussion should be done to explain the stabilization of the system by means of (classical) MD simulations.
- 5) On the Fig. 4, it has been suggesting that His153 act as base to remove a proton from NH₃⁺ of peptide. However, this involves a pKa shift for imidazole and amine groups. How imidazole group of His153 is stabilized during the next steps (once, just on last step it will be neutralized)? Could any amino-acid residue be involved?
- 6) On the Fig. 4, a water molecule (P state) is responsible to neutralize His153. What is the thermodynamic cost to place it there, once it has not been shown on previous steps of proposed mechanism?
- 7) For the 1st step of proposed mechanism, a water molecule could remove a proton from NH₃⁺ of peptide. Then, I suggest include a QM/MM calculation using a water molecule as proton abstractor instead of His153.

Reviewer #4 (Remarks to the Author):

The manuscript from Caffrey and colleagues is an outstanding detail of experimental and computational approaches to understand a representative member of the Lit family of transacylases. The document is well written, the experimental data are nicely done and the conclusions are largely supported by the data. I would recommend a few editorial changes that would improve the manuscript appeal to a broader audience.

1. The Discussion section is overtly long and includes many lines of thought that are not supported experimentally. Please condense and leave off frivolous items.
2. Abstract line 26: "Here we report the crystal structure of the enzyme from.." Which enzyme? Lit should be explicitly stated here.
3. Table S2 is nice but can the authors include an actual evolutionary tree in the Supplemental section? It would help convey relationship amongst other related members.
4. Figure S7. Please state in the legends how many replicated were conducted for each point since error bars are shown.
5. Methods section: please state the criteria used to determine the resolution cutoffs for the structures. The crystallographic table may not make sense to the non-experts.
6. Figure 3E should really show a difference Fourier map superimposed on the monoolein. This is an

important figure and the audience should have some confidence that the density for the ligand is respectable enough to support the distances that are shown.

7. Line 785 in the Methods suggest that Sequence Logos were generated but I could not find them in the manuscript. Please delete if this is extraneous.

Response to REVIEWER COMMENTS

Reviewer #1 (Remarks to the Author):

This research article presented structure of lipoprotein intramolecular transacylase (Lit) in Gram-positive bacteria and discussed the potential reaction mechanism. The authors presented several crystal structures of the Lit, a novel enzyme in modifying bacterial lipoproteins. The authors also coupled the study with functional assays, chemical biology, mutagenesis, and computational analyses to provide more insight to Lit functions. This was a well-thought out study providing lots of insight into Lit functionality and mode of action, and the authors concluded with proposed mechanisms based upon all the evidence presented. The study should provide an additional tool in understanding bacterial pathogenesis and evasion of host immunity.

The crystal structures provided insight into the possible binding modes of the endogenous ligand by way of the two monoolein (MO) molecules. Unfortunately due to the unreleased nature of the PDB coordinates, it would have been great to be able to see the 2Fo-Fc map around MO molecules in each PDB molecule. The authors showed partial views in Fig.S18. In Fig.S18B the density for FP2 is incomplete, unfortunately. Would it be more beneficial also for the authors to present the 2Fo-Fc maps of the active sites from Lit_2 structure (1.94 Å structure)?

The requested maps have been included in the revised manuscript. See Supplementary Figure 6.

Crystallographic table details for PDB:7B00 (Lit_1) showed that CC1/2 is 0.96 (0.18) - is this a typo?

The corrections have been made in Supplementary Table 1. We thank the Reviewer for bringing these to our attention.

Line 128: two structures are solved and are they look highly similar? If so, what is the RMSD?

RMSD values have been included in the revised manuscript under Lit structure. They range from 0.21 to 0.40 over 211 to 216 residues.

An interesting method that the authors brought up is the use of the cubicon method for concentrating H85A-peptide mutant to eventually set up crystallization trials.

Indeed, the cubicon method proved invaluable.

Overall this is a novel enzyme and the authors utilized many different methods to elucidate its mechanism of action. Taken into context within the

whole system of lipoprotein modifying enzymes, this study represents a breakthrough in the potential to target bacterial pathogens.

We thank the Reviewer for these very favourable comments.

Reviewer #2 (Remarks to the Author):

I have read the manuscript "Structural basis of the membrane intramolecular transacylase reaction responsible for lyso-form lipoprotein synthesis" by Olatunji, S. et al that has been submitted for consideration for publication in Nature Communications. The study reports the crystal structure with biochemical characterization of Lit, an enzyme that remodels lipoproteins. Lipoproteins are ubiquitous components of bacterial membranes that play central roles in both bacterial physiology and infection- as such, the report should appeal to a wide swath of the readership. Lit is also a unique enzyme with respect to activity and structure, as prior insights regarding enzymes catalyzing intramolecular transacylations are rare. Examples of 8-member ring intermediates are equally rare- thermodynamic computation/simulations and in vitro assays provide some support for this intermediates though its existence must remain qualified. Oncemore, this is a timely report given the rapid advancements in the field. Overall the paper is well written, logical to follow, and corroborates/significantly advances prior knowledge regarding this enzyme.

The introduction needs clarifications/corrections as suggested below, some of the speculation regarding specific residues can be trimmed, but otherwise, I am enthusiastic about the manuscript.

Specifics Comments/Suggestions:

1. Abstract, line 23-24: Counting lipoprotein enzyme targets is more complex than implied with this sentence as Meredith/Gotz labs recently reported another N-acylation system in Staphylococcus aureus LnsAB mBio. 2020 Jul 28;11(4):e01619-20. doi: 10.1128/mBio.01619-20). Also should Lol lipoprotein (LP) transport system be counted here? Please update and/or rework this sentence.

We agree with the Reviewer and have reworked the sentence. Reference to Lol is not appropriate here.

2. line 65- "dagylated"- terminology here and throughout is slang

'Dagylated' has been replaced by 'diacylglyceryl-modified' throughout the revised manuscript.

3. line 66-67- This passage is confusing- all LPs in gram negative (at least model E coli) are matured by Lnt to form TA-LP, regardless of whether they are destined for retention in inner or eventually transported to outer membrane. While they must be triacylated for efficient recognition by Lol, the peptide/exclusion signal downstream of Cys is thought to determine further trafficking to OM. Please revise.

The text has been revised as suggested by the Reviewer.

4. Fig 1: While overall TLR2/1/6 signaling is indeed extremely weak for lyso-LP, neutralizing TLR1 or TLR6 antibody studies, and TLR2 or 1 specific deletion NF-kB reporter HEK strain assays actually suggested residual lyso-LP induced signaling more likely is from TLR2/6 than TLR2/1 (J Bacteriol 2019 Jun 10;201(13):e00195-19. doi: 10.1128/JB.00195-19). Please adjust figure to indicate (i.e. “weak TLR2” signaling for Lit)- NOTE: this is later covered in Discussion section, but figure should still be adjusted to match text

Figure 1 has been corrected. We thank the Reviewer for carefully reviewing our manuscript and catching this typo.

5. line 69- indicate which acyl chain of PE is utilized by Lnt

The acyl chain transferred is identified in the revised text.

6. line 82-84: Ref 24, 25 Classifying these as BLPs introduces unnecessary confusion as the (already established) generic “lipoprotein” nomenclature is unfortunate. As noted by authors, the references describe protein acylation on an N-terminal Gly substrate, that are not processed by Lgt, that do not have a diacyl-glycerol modified Cys, no canonical lipobox signal, and are SpsI (not SpsII) dependent. Lumping together with canonical “BLP” as discussed in manuscript is awkward. Consider removing for clarity.

We appreciate and share the concern of the Reviewer. It is for this reason we chose the title for Figure 1 which refers specifically to cysteine-based modifications. The last sentence in the legend (lines 82-84, original manuscript) is there simply to alert the Reader to the fact that other types of lipoproteins exist in bacteria.

7. line 201-203: Many of these (analogous) residues, including the two His and aromatic residues, have been shown to be functionally important for Lit from *E. faecalis* in vivo using an alanine scan library (Ref 1 in paper)- how do those reported here compare?

Highly conserved/functionally important residues identified in Lit from *B. cereus* on the basis of sequence and mutational analysis in this study (Figure 3) include:

Leu59, Phe86, His85, Lys90, Val89, Phe149, His153, Phe157, and Trp162

The residues identified as potentially functionally important in Lit from *E. faecalis* by Ala scanning in Reference 1 include:

His89(85), Phe90, Phe149, Phe153(149), Phe156, His157(153), Phe161(157), Trp166(162), Phe168, and Phe184 – *B. cereus* numbering in brackets.

8. line 382-383/409-410- Mass spec evidence has been provided for which of the two acyl chains is transferred (i.e. sn2) using “mixed chain” DAG (unsaturated Sn2 and saturated Sn1 acyl groups) with isolated membranes in vitro, albeit cyclic transfer verse acyl-enzyme is unclear

This is covered in the manuscript in the following statement (lines 408-410 in the original manuscript: “A similar conclusion was reached by Armbruster et al. (1) on the basis of MS analysis of a recombinantly produced lipoprotein containing deuterated acyl chains in the DAG moiety.”

9. Paragraph beginning at Line 503, Lyso-PE: the text here reads quite similar to that which has already been proposed in the discussion of reference 1 and should be cited

Reference 1 was in early versions of the manuscript and, with reworking, got omitted. It is in the revised manuscript. We thank the Reviewer to pointing this out.

10. line 559- “pumped” copper?

We use ‘piped’ as in conveyed or channelled or transported.

11. line 561-577- This is an awkward way to end the paper- I would suggest trimming as the more pertinent aspects can be incorporated earlier in the mechanism part of text where the unique 8-member intermediate is proposed and first discussed

Insights into the reaction mechanism of the Lit enzyme are an important outcome of this study that are worthy of discussion. We agree with the Reviewer that the paragraph on 8-membered intermediates is perhaps a somewhat abrupt ending to the discussion. Accordingly, we have relocated it and ended the discussion with the paragraph on TLR interaction.

Reviewer #3 (Remarks to the Author):

Dear Editor,

Olatunji and co-workers report the crystal structure of lipoprotein from *Bacillus cereus* and couple it with functional assays, mutagenesis, chemical biology, and computational analysis to establish its mechanism of action. This is a well-conducted study and suitable to be published on the *Nature Communications*.

In general, the study is very interesting; however, I have the following concerns on the manuscript which should be addressed before publication

1) The authors should provide in the introduction a critical comparison between all four lipoproteins, highlighting major differences

We assume the Reviewer is referring to the four posttranslational modifying enzymes. The reactions they catalyse are described in the Introduction and in Figure 1.

2) A potential energy surface (PES) obtained from QM/MM calculations for

the whole studied mechanism should be included on the main text of the manuscript.

We thank the Reviewer for the suggestion. However, we believe the PES plots are better left in the SI because this is where the two proposed reaction mechanisms are compared (Supplementary Figure 15). A perusal of the data presented side-by-side in this figure enables the Reader to conclude that the acid-base reaction mechanism is feasible while the alternative enzyme acylation mechanism is not.

3) A Table containing relevant geometric parameters (atomic distances) during QM/MM PES should be provide on the main text of the manuscript and related to their respective states.

The relevant atomic distances from the QM/MM study as they relate to their respective states are shown in Figure 4. We believe that inclusion of interatomic distances in the main text figure provides the Reader with a more intuitive understanding of the reaction mechanisms. Nevertheless, as requested by the Reviewer, we have included Supplementary Table 5 with the relevant distance information - see Supplementary Table 5 of the revised manuscript which is referred to in the main text.

4) As a total of 500 ns of MD was performed to system, I suggest a free energy landscape (FEL) from the principal component analysis (PCA) to avail the most favourable representative (more stable) structure. Finally, a suitable discussion should be done to explain the stabilization of the system by means of (classical) MD simulations.

The suggestion to build up the free energy landscape from the PCA analysis is a good one and we thank the Reviewer for it. However, it would not reveal new information on the conformational dynamics and stability of Lit. Our RMSD analysis of the simulated MD trajectories shows that the simulations are stable and deviate little from the crystal structure. The C α atom cluster analysis of the equilibrated MD trajectories shows that there were no major conformational changes and identified the stable conformations of Lit for use in the subsequent QM/MM study. We have commented on the stability of the Lit enzyme in the classical MD simulations at the end of the 'Lit structure' section and have supplemented this with Supplementary Figure 3 and Supplementary Movie 1. PCA analysis (Supplementary Figure 22) was done primarily to show the directions of principal protein motions and to suggest a pathway into and out of the active sites.

5) On the Fig. 4, it has been suggesting that His153 act as base to remove a proton from NH₃⁺ of peptide. However, this involves a pKa shift for imidazole and amine groups. How imidazole group of His153 is stabilized during the next steps (once, just on last step it will be neutralized)? Could any amino-acid residue be involved?

We thank the Reviewer for this comment. After the proton is abstracted by His153 from the N-terminal amino group, the histidine side chain undergoes a conformer flip and moves away from the active site. In this new conformation, the positively charged imidazole ring is stabilised

by cation- π interaction with Trp162. This is commented on in the 'Reaction mechanism' section of the Results and is nicely illustrated in Supplementary Movie 6. The interaction with Trp162 stabilises His153 before it loses its proton to the bulk solvent in preparation for the next round of catalysis.

6) On the Fig. 4, a water molecule (P state) is responsible to neutralize His153. What is the thermodynamic cost to place it there, once it has not been shown on previous steps of proposed mechanism?

The side chain of His153 adopts a new conformation after the initial proton abstraction from the N-terminal amino group of the lipopeptide substrate. In this state, the side chain is solvent exposed as evidenced by MD simulations (Supplementary Movie 6). For this reason, we chose not to place any water molecules in the vicinity of His153. As the deprotonation of His153 is a part of the enzyme regeneration mechanism, we simply assigned nearby water molecules to the QM region of the QM/MM calculations and studied proton transfer there. There was no additional thermodynamic cost associated with the deprotonation of His153 to its neutral state for the subsequent round of catalysis.

7) For the 1st step of proposed mechanism, a water molecule could remove a proton from NH_3^+ of peptide. Then, I suggest include a QM/MM calculation using a water molecule as proton abstractor instead of His153.

Indeed, the water molecule can abstract a proton from the amino group of the lipopeptide substrate as suggested by the Reviewer. To explore this option, we performed a QM/MM study in which nearby water molecules were added into the QM region. The potential energy scan (Figure 1 below) shows the proton transfer from the NH_3^+ group to a water molecule with the activation energy barrier of ~ 15.7 kcal/mol. This finding indicates that proton abstraction by a water molecule is possible. However, it is far less favourable in comparison to proton abstraction by His153, where the activation energy barrier is only 1.36 kcal/mol (Supplementary Figure 15).

Figure 1. The QM/MM potential energy scan for the abstraction by a water molecule of a proton from the N-terminal ammonium group of the substrate lipopeptide. The distance on the x-axis refers to the separation between the oxygen atom of water and the hydrogen atom of the ammonium group.

Reviewer #4 (Remarks to the Author):

The manuscript from Caffrey and colleagues is an outstanding detail of experimental and computational approaches to understand a representative member of the Lit family of transacylases. The document is well written, the experimental data are nicely done and the conclusions are largely supported by the data. I would recommend a few editorial changes that would improve the manuscript appeal to a broader audience.

1. The Discussion section is overtly long and includes many lines of thought that are not supported experimentally. Please condense and leave off frivolous items.

From the authors' perspective, the items included in the Discussion section are important and should be included in the final manuscript. It is agreed that some of the discussion is speculative. However, this is the purpose of having a Discussion section. All speculation in the Discussion is supported by well-reasoned arguments.

2. Abstract line 26: "Here we report the crystal structure of the enzyme from.." Which enzyme? Lit should be explicitly stated here.

Lit has been added as suggested.

3. Table S2 is nice but can the authors include an actual evolutionary tree in the Supplemental section? It would help convey relationship amongst other related members.

A phylogenetic tree has been included in the revised manuscript (Supplementary Figure 5) as suggested by the Reviewer.

4. Figure S7. Please state in the legends how many replicated were conducted for each point since error bars are shown.

The number of replicates is two. This information is now included in Supplementary Figure 9.

5. Methods section: please state the criteria used to determine the resolution cutoffs for the structures. The crystallographic table may not make sense to the non-experts.

Cut-off criteria have been spelled out under Methods (Diffraction data collection and processing) as suggested by the Reviewer. Additionally, a few minor corrections have been made to the Crystallographic Supplementary Table 1.

6. Figure 3E should really show a difference Fourier map superimposed on the monoolein. This is an important figure and the audience should have some confidence that the density for the ligand is respectable enough to support the distances that are shown.

Electron densities that define the monoolein molecules in the binding pocket of all structures included in this manuscript are shown in Supplementary Figure 6.

7. Line 785 in the Methods suggest that Sequence Logos were generated but I could not find them in the manuscript. Please delete if this is extraneous.

This typo in the original submission has been corrected. WebLogo has been changed to Sequence logo. We thank the Reviewer for pointing out the error.

REVIEWERS' COMMENTS

Reviewer #1 (Remarks to the Author):

This revised manuscript addressed all of my concerns and is ready to be published at Nature Communications.

Reviewer #2 (Remarks to the Author):

All of my concerns have been addressed adequately.

Reviewer #3 (Remarks to the Author):

Dear Editor,

Olatunji and co-workers report the crystal structure of lipoprotein from *Bacillus cereus* and couple it with functional assays, mutagenesis, chemical biology, and computational analysis to establish its mechanism of action.

This is a well-conducted study and suitable to be published on Nature Communications.

In this new version of the manuscript, the authors have answered and attended to my previous concerns. Then, I can suggest this manuscript for publication.

Reviewer #4 (Remarks to the Author):

The authors have addressed all of the concerns raised during the initial review of this manuscript. I heartily endorse publication of this very elegant story and look forward to seeing the final version in print in Nature Communications.

A response to the Reviewers' comments (copied below) is not needed other than to thank them all sincerely for the care and attention they devoted to reviewing our manuscript.

REVIEWERS' COMMENTS

Reviewer #1 (Remarks to the Author):

This revised manuscript addressed all of my concerns and is ready to be published at Nature Communications.

Reviewer #2 (Remarks to the Author):

All of my concerns have been addressed adequately.

Reviewer #3 (Remarks to the Author):

Dear Editor,

Olatunji and co-workers report the crystal structure of lipoprotein from *Bacillus cereus* and couple it with functional assays, mutagenesis, chemical biology, and computational analysis to establish its mechanism of action. This is a well-conducted study and suitable to be published on Nature Communications.

In this new version of the manuscript, the authors have answered and attended to my previous concerns. Then, I can suggest this manuscript for publication.

Reviewer #4 (Remarks to the Author):

The authors have addressed all of the concerns raised during the initial review of this manuscript. I heartily endorse publication of this very elegant story and look forward to seeing the final version in print in Nature Communications.